



# An empirical algorithm to map perennial firn aquifers, ice slabs, and perched firn aquifers within the Greenland Ice Sheet using satellite L-band microwave radiometry

Julie Z. Miller[1,2], Riley Culberg[3], David G. Long[4], Christopher A. Shuman[5],
Dustin M. Schroeder[3,6], Mary J. Brodzik[1,7]

[1] Cooperative Institute for Research in Environmental Sciences, University of Colorado, Boulder, Colorado, USA
[2] Earth Science and Observation Center, University of Colorado, Boulder, Colorado, USA
[3] Department of Electrical Engineering, Stanford University, Stanford, California, USA
[4] Department of Electrical and Computer Engineering, Brigham Young University, Provo, Utah, USA
[5] University of Maryland, Baltimore County, Joint Center for Earth Systems Technology at Code 615,
Cryospheric Sciences Laboratory NASA Goddard Space Flight Center, Greenbelt, Maryland, USA
[6] Department of Geophysics, Stanford University, Stanford, CA, USA
[7] National Snow and Ice Data Center, University of Colorado, Boulder, Colorado, USA
Correspondence to: jzmiller.research@gmail.com

## Abstract

*Perennial firn aquifers are subsurface meltwater reservoirs formed from a water-saturated firn layer. They have been observed within the percolation facies of glaciated regions experiencing intense seasonal surface melting and high snow accumulation. Widespread perennial firn aquifers have been identified within the Greenland Ice Sheet (GrIS) via field expeditions, airborne ice-penetrating radar surveys, and satellite microwave sensors. In contrast, ice slabs are nearly-continuous ice layers that form on spatial scales of kilometers as a result of surface and subsurface water-saturated snow and firn layers sequentially refreezing following multiple melting seasons. They have been observed within the percolation facies of glaciated regions experiencing intense seasonal surface melting, but in areas where snow accumulation is at least ~25% lower as compared to perennial firn aquifer areas. Widespread ice slabs have recently been identified within the GrIS via field expeditions and airborne ice-penetrating radar surveys, specifically in areas where perennial firn aquifers typically do not form. However, ice slabs have yet to be inferred from space. Together, these two ice sheet features represent distinct, but related, sub-facies within the broader percolation facies of the GrIS that can be defined primarily by differences in snow accumulation, which influences the englacial hydrology and thermal characteristics of firn layers at depth.*

*Here, for the first time, we use enhanced-resolution vertically-polarized L-band brightness temperature ($T_V^B$) imagery (2015-2019) generated using observations collected over the GrIS by NASA's Soil Moisture Active Passive (SMAP) satellite to map both perennial firn aquifer and ice slab areas as a continuous system over the percolation facies. We also map 'perched' firn aquifer areas, which we define as areas where shallow water-saturated firn layers transiently form on top of buried ice slabs, or other semi-impermeable layers within the snow and firn. An empirical algorithm*



*previously developed to map the extent of Greenland's perennial firn aquifers via fitting*
*exponentially decreasing temporal L-band signatures to a set of sigmoidal curves is recalibrated to*
*also map the extent of ice slab and perched firn aquifer areas using airborne ice-penetrating radar*
*surveys collected by NASA's Operation Ice Bridge (OIB) campaigns (2010-2017). Our SMAP-derived*
*maps show that between 2015 and 2019, perennial firn aquifer areas extended over ~64,000 km², ice*
*slab areas extended over ~76,000 km², and perched firn aquifer areas extended over ~37,000 km².*
*Combined together, these three sub-facies are the equivalent of ~24% of the percolation facies of*
*the GrIS. As Greenland's climate continues to warm, and seasonal surface melting increases in*
*extent, intensity, and duration, quantifying the possible rapid expansion of each of these sub-facies*
*using satellite L-band microwave radiometry has significant implications for understanding ice*
*sheet-wide variability in englacial firn hydrology resulting in meltwater-induced hydrofracturing and*
*accelerated ice flow as well as high-elevation run-off that can impact the mass balance and stability*
*of the GrIS.*

## 1     Introduction

The recent launch of several satellite L-band microwave radiometry missions by NASA (Aquarius mission,
Levine, et al., 2007; Soil Moisture Active Passive (SMAP) mission, Entekhabi et al., 2010) and ESA (Soil
Moisture and Ocean Salinity (SMOS), Kerr et al., 2010) has provided a new Earth-observation tool
capable of detecting stored meltwater ~tens of meters to kilometers beneath the surface of ice sheets.
Jezek et al. (2015) recently demonstrated that in the high-elevation (~3500 m.a.s.l.) dry snow facies of the
Antarctic Ice Sheet, meltwater stored in subglacial Lake Vostok can be detected ~4000 m beneath the ice
sheet surface. Subglacial lakes represent radiometrically cold subsurface meltwater reservoirs. Upwelling
L-band emissions from the radiometrically warm bedrock underlying the subglacial lakes are effectively
blocked by high reflectivity and attenuation at the interface between bedrock and the overlying lake bottom.
This results in a lower observed microwave brightness temperature ($T^B$) at the ice sheet surface as
compared to other dry snow facies areas where bedrock contributes to L-band emissions depth-integrated
over the entire ice sheet thickness.
Similar to subglacial lakes, perennial firn aquifers also represent radiometrically cold subsurface
meltwater reservoirs (Miller et al., 2020) formed from a ~4 m-25 m thick water-saturated firn layer (Koenig
et al., 2014; Montgomery et al., 2017; Chu et al., 2018). They have been observed via field expeditions and
airborne ice-penetrating radar surveys in the lower-elevation (< ~2000 m.a.s.l.) percolation facies of the
Greenland Ice Sheet (GrIS), at depths from between ~1 m and 40 m beneath the ice sheet surface (Miège
et al. 2016), and in areas that experience intense seasonal surface melting (>650 mm yr⁻¹) during the
melting season and high snow accumulation (>800 mm yr⁻¹) during the freezing season (Forster et al.,
2014). High snow accumulation in perennial firn aquifer areas thermally insulates water-saturated firn layers
from the cold atmosphere allowing seasonal meltwater to be stored in liquid form if the overlying seasonal
snow layer is sufficiently thick (Kuipers Munneke et al., 2014). Koenig et al. (2014) estimated that the





volumetric fraction of meltwater stored within the pore space of Greenland's perennial firn aquifers just prior
to melt onset ranges from between ~10% and 25%, which limits the upward propagation of electromagnetic
energy from greater depths within the ice sheet. Large volumetric fractions of meltwater within the firn pore
space results in high reflectivity and attenuation at the interface between water-saturated firn layers and
the overlying refrozen firn layers, and between glacial ice or a semi-impermeable layer and the overlying
water-saturated firn layers. Upwelling L-band emissions from deeper glacial ice and the underlying bedrock
are effectively blocked.

While perennial firn aquifers are radiometrically cold, the slow refreezing of deeper firn layers

saturated with large volumetric fractions of meltwater represents a significant source of latent heat that is
continuously released throughout the freezing season. Refreezing of seasonal meltwater by the descending
winter cold wave (Pfeffer et al., 1991), and the subsequent formation of embedded ice structures (i.e.,
horizontally-oriented ice layers and ice lenses, and vertically-oriented ice pipes; Benson et al., 1960;
Humphrey et al., 2012; Harper et al., 2012) within the upper snow and firn layers represents a secondary
source of latent heat. These heat sources help maintain meltwater at depth. Perennial firn aquifer areas
are radiometrically warmer than other percolation facies areas where the single source of latent heat is via
refreezing of seasonal meltwater. This results in a higher observed $T^B$ at the ice sheet surface during the
freezing season as compared to other percolation facies areas where seasonal meltwater is fully refrozen
and stored exclusively as embedded ice.

Many open questions remain about Greenland's perennial firn aquifers, regarding initial formation,

extent, depth, flow characteristics, timescales of refreezing and/or englacial drainage, and connections to
the subglacial hydrological system. Seasonal surface melting over the GrIS has increased in extent,
intensity, and duration since the beginning of the satellite era (Steffen et al., 2004; Tedesco e al., 2008;
Tedesco et al., 2011; Nghiem et al., 2012; Tedesco et al., 2016; Tedesco and Fettweis, 2020; Cullather et
al., 2020). If this trend continues (Franco et al., 2013; Noël et al., 2021), subsequent increases in the
volume of meltwater stored within Greenland's perennial firn aquifers will increase the possibility of
crevasse-deepening via meltwater-induced hydrofracturing (Alley et al., 2005; van der Veen, 2007),
especially if crevasse fields laterally expand into perennial firn aquifer areas as a result of accelerated ice
flow (Colgan et al., 2016). Meltwater-induced hydrofracturing is an important component of supraglacial
lake drainage during the melting season (Das et al., 2008; Stevens et al., 2015) leading to at least temporary
accelerated flow velocities (Zwally et al., 2002; Joughin et al., 2013; Moon et al., 2014) and mass balance
changes (Joughin et al., 2008). Greenland's firn perennial aquifers may also support meltwater-induced
hydrofracturing, even during the freezing season (Poinar et al., 2017; 2019).

Recently, mapping the extent of Greenland's perennial firn aquifers from space was demonstrated

using satellite L-band microwave radiometry (Miller et al., 2020). Exponentially decreasing temporal L-band
signatures observed in enhanced-resolution vertically-polarized L-band brightness temperature ($T_V^B$)
imagery (2015-2016) generated using observations collected over the GrIS by the microwave radiometer
on the SMAP satellite (Brodzik et al., 2019) were correlated with a single year of perennial firn aquifer





detections (2016) identified via the Center for Remote Sensing of Ice Sheets (CReSIS) Multi-Channel
Coherent Radar Depth Sounder (MCoRDS) flown by NASA's Operation Ice Bridge (OIB) campaigns (Miège
et al. 2016; Rodriguez-Morales et al, 2014). An empirical algorithm to map extent was developed by fitting
temporal L-band signatures to a set of sigmoidal curves derived from the continuous logistic model.
The relationship between the radiometric, and thus the physical, temperature of perennial firn
aquifer areas, as compared to other percolation facies areas, forms the basis of the empirical algorithm.
Miller et al. (2020) hypothesized that the dominant control on the relatively slow exponential rate of $T_V^B$
decrease over perennial firn aquifer areas is physical temperature versus depth. L-band emissions from
the radiometrically warm upper snow and firn layers decrease during the freezing season as embedded ice
structures slowly refreeze at increased depths below the ice sheet surface. In the percolation facies,
refreezing of seasonal meltwater results in the formation of an intricate network of embedded ice structures
that are large (~10-100 cm long, ~10-20 cm wide; Jezek et al., 1994) relative to the L-band wavelength
(~21 cm). Embedded ice structures induce strong volume scattering (Rignot et al., 1993; Rignot 1995) that
decreases $T^B$ (Zwally, 1977; Swift et al. 1985; Jezek et al., 2018).
Ice slabs are ~1 m–16 m thick nearly-continuous ice layers that form on spatial scales of kilometers
as a result of surface and subsurface water-saturated snow and firn layers sequentially refreezing following
multiple melting seasons (Machguth et al., 2016; McFerrin et al., 2019). Over time, they become dense low-
permeability solid-ice layers overlying deeper permeable firn layers. Similar to perennial firn aquifers, ice
slabs have been observed via field expeditions and ice-penetrating airborne radar surveys in the lower-
elevation (< ~2000 m.a.s.l.) percolation facies of the GrIS. They form at depths from between ~1 m and 20
m beneath the ice sheet surface. Particularly in areas that experience intense seasonal surface melting
(>600 mm yr−1) during the melting season, and lower snow accumulation (<600 mm yr−1) during the
freezing season as compared to perennial firn aquifer areas (McFerrin et al., 2019). Lower snow
accumulation in ice slab areas results in a seasonal snow layer that is insufficiently thick to thermally
insulate water-saturated firn layers and seasonal meltwater is instead stored as embedded ice. Refreezing
of seasonal meltwater by the descending winter cold wave, and the subsequent formation of ice slabs as
well as other embedded ice structures within the upper snow and firn layers is the single source of latent
heat in ice slab areas. While ice slab areas are radiometrically warmer than other percolation facies areas
with a lower volumetric fraction of embedded ice, they are radiometrically colder than perennial firn aquifer
areas. This results in a lower observed $T^B$ at the ice sheet surface during the freezing season.
Consistent with recent seasonal surface melting trends, meltwater run-off has accelerated to
become the dominant mass loss mechanism over the GrIS (van den Broeke et al., 2016). However,
significant uncertainty remains in meltwater run-off estimates in the percolation facies as a result of the lack
of knowledge of heterogeneous infiltration processes within the snow and firn layers (Pfeffer and Humphrey,
1996), the depths to which meltwater can descend beneath the ice sheet surface (Humphrey et al., 2012),
and the formation of englacial firn hydrological features (Benson et al., 1960; Humphrey et al., 2012; Forster
et al., 2014), especially ice layers and ice slabs (Machguth et al., 2016, McFerrin et al., 2019; Culberg et





al., 2021). A notable example of this lack of knowledge is the identification by Forster et al., (2014) of
widespread perennial firn aquifers within the percolation facies of the GrIS via airborne ice-penetrating radar
surveys collected by NASA's OIB campaigns (2010-2014; Rodriguez-Morales et al, 2014) that store large
volumes (~140 Gt; Koenig et al., 2014) of meltwater that was previously unknown. The mapped extent
(2010-2014) shown in Forster et al., (2014) can be distinctly observed in 1978 enhanced resolution Ku-
band radar backscatter imagery (Long and Drinkwater, 1994) collected by the radar scatterometer on
NASA's first Earth-observing satellite - the Seasat-A mission (Jones et al., 1982). This suggests that
Greenland's perennial firn aquifers have likely existed undetected in the deeper firn layers of the percolation
facies for decades. Meltwater storage in both solid (i.e., embedded ice structures) and liquid (i.e., perennial
firn and perched firn aquifers) form can buffer meltwater run-off in the percolation facies (Harper et al.,
2012). However, the formation of near-surface ice layers and ice slabs reduces the pore space within the
upper snow and firn layers and facilitates lateral meltwater flow with minimum vertical percolation into the
deeper firn layers, thus enhancing meltwater run-off downslope towards the periphery. Lateral meltwater
flow across ice layers overlying deeper permeable firn layers was first postulated by Müller (1962). The
theory was then further developed by Pfeffer et al., (1991) as an end-member case for meltwater run-off,
with the other end member case being lateral meltwater flow across superimposed ice in the wet snow
facies and/or across glacial ice in the ablation facies. McFerrin et al., (2019) recently identified widespread
near-surface ice slabs within the percolation facies of the GrIS via airborne ice-penetrating radar surveys
collected by NASA's OIB campaigns (2010-2014; Rodriguez-Morales et al, 2014). Lateral meltwater flow
and high-elevation (~1850 m.a.s.l) meltwater run-off across the identified ice slabs was also observed in
visible satellite imagery collected by the NASA-USGS Landsat 7 mission (e.g. Goward et al., 2001). This
was also observed during the anomalous 2012 melting season (McFerrin et al., 2019) during which
seasonal surface melting extended over ~99% of the GrIS (Nghiem et al., 2012)

In this study, we use enhanced-resolution L-band $T_V^B$ imagery (2015-2019) generated using
observations collected over the GrIS by the microwave radiometer on the SMAP satellite (Brodzik et al.,
2019) to map ice sheet-wide englacial firn hydrological features within the percolation facies. First, we adapt
our empirical algorithm to map the extent of Greenland's perennial firn aquifers (Miller et al., 2020). We
correlate exponentially decreasing temporal L-band signatures with five years of perennial firn aquifer
detections (2010-2014) identified via the CReSIS Accumulation Radar (AR) flown by NASA's OIB
campaigns (Miège et al. 2016), and three years of additional detections (2015-2017) more recently
identified via MCoRDS (Miller et al., 2020). Next, we extend our empirical algorithm to also map the extent
of ice slab and perched firn aquifer areas. We identify distinct temporal L-band signatures in $T_V^B$ time series
over ice slab detections (2010-2014) recently identified via AR (McFerrin et al., 2019). Similar to temporal
L-band signatures over perennial firn aquifer areas, temporal L-band signatures over ice slab areas are
exponentially decreasing during the freezing season, however, the rate of $T_V^B$ decrease is slightly more
rapid. We correlate these relatively rapidly exponentially decreasing temporal L-band signatures with five
years of AR-derived ice slab detections. Additionally, we correlate exponentially decreasing temporal L-



band signatures with AR- and MCoRDS-derived detections where perennial firn aquifer and ice slab areas
overlap. We identify these transitional areas as perched firn aquifer areas. We infer that, in these areas,
shallow water-saturated firn layers transiently form on top of buried ice slabs or other semi-impermeable
layers, such as spatially coherent melt layers that form in the higher elevations (> ~2000 m.a.s.l.) of the
percolation facies and the dry snow facies that were recently identified via AR (Culberg et al., 2021).
Perched firn aquifers likely form during some melting seasons as a result of interannual variability in surface
melting and snow accumulation, and the formation of englacial firn hydrological features. Finally, we re-
calibrate the sigmoidal curves to map the extent of perennial firn aquifer, ice slab, and perched firn aquifer
areas over the percolation facies of the GrIS

## 2    Methods
### 2.1    The Soil Moisture Active Passive (SMAP) Mission
The key science objectives of NASA's SMAP mission (https://smap.jpl.nasa.gov/) are to map terrestrial soil
moisture and freeze/thaw state over Earth's land surfaces from space. However, the global L-band $T^B$
observations collected by the SMAP satellite also have many cryospheric applications. Mapping ice sheet-
wide englacial firn hydrological features over Earth's polar ice sheets represents an interesting analog and
an innovative extension of the science objectives. Measurements of moisture (i.e., defined in this study in
terms of the volumetric fraction of meltwater within the upper snow and firn layers of the percolation facies)
and freeze-thaw state (i.e., defined in this study in terms of the firn saturation parameter (see Section 2.4.3)
and the refreezing rate parameter (see Section 2.4.4)) are critical to understanding the hydrospheric state
over Earth's polar ice sheets. Perennial firn aquifers, ice slabs, and perched firn aquifers represent recently
identified components of the hydrosphere that are capable of storing large volumes of meltwater in both
solid and liquid form that can initiate meltwater-induced hydrofracturing and accelerated ice flow as well as
high-elevation run-off, and impact the mass balance and stability of the GrIS. Critically, the majority of
meltwater is stored at depths that only L-band satellite microwave sensors (i.e., radiometers, radar
scatterometers, and synthetic aperture radars) are capable of detecting.

Previous and current satellite microwave radiometer, radar scatterometer, and synthetic aperture
radar missions that operate in the frequency range between 37 GHz (Ka-band) and 5.3 GHz (C-band) have
provided a multi-decadal (1978-present) record of multi-frequency $T^B$ and radar backscatter observations
over Earth's polar ice sheets since the beginning of the satellite era. The most common geophysical
parameter mapped over ice sheets using these observations is the extent of seasonal surface melting. The
key difference between L-band and higher frequency satellite microwave sensors is penetration depth.
When the snow and firn layers are saturated with meltwater during the melting season, the penetration
depth of both L-band and higher frequency satellite microwave sensors is less than ~a meter. When surface
and subsurface water-saturated snow and firn layers and embedded ice structures subsequently refreeze,
the penetration depth of higher frequency satellite microwave sensors ranges from between ~centimeters
and meters. During the freezing season, water-saturated snow and firn layers either completely refreeze





(i.e., ice layers, ice slabs, spatially coherent melt layers) or underlay the refrozen upper snow and firn layers
of the percolation facies and descend to depths ranging from between ~1 m and 40 m (Miège et al., 2016)
beneath the ice sheet surface (i.e., perennial and perched firn aquifers). While the upper surface of stored
meltwater in some perennial and perched firn aquifers may remain at depths that are shallow enough to be
directly detected by C-band satellite microwave sensors, the mean depth just prior to melt onset (~22 m;
Miège et al., 2016) is too deep to be detected at this wavelength. L-band satellite microwave sensors can
detect perennial firn aquifers from as much as an order of magnitude deeper than can be observed by C-
band satellite microwave radiometers. Deep enough to directly detect the upper surface of stored meltwater
over the entire depth range mapped by airborne ice-penetrating radar surveys over the GrIS.
## 2.1    SMAP Enhanced-Resolution L-band $T^B$ Imagery

NASA's SMAP satellite was launched 31 January 2015 and carries a microwave radiometer that operates
at a frequency of 1.41 GHz (L-band) (Enkentabi et al., 2010). It is currently collecting observations of
vertically and horizontally-polarized $T^B$ over Greenland. The surface incidence angle is ~40°, and the
radiometric accuracy is ~1.3 K (Piepmeier et al., 2017).
The Scatterometer Image Reconstruction (SIR) algorithm was developed to reconstruct coarse
resolution satellite scatterometry imagery on a higher spatial resolution grid (Long et al., 1993; Early and
Long, 2001). The SIR algorithm has been adapted for coarse resolution satellite microwave radiometry
imagery (Long and Daum, 1998; Long and Brodzik, 2016; Long et al., 2019). The microwave radiometer
form of the SIR algorithm (rSIR) exploits the measurement response function (MRF) for each observation,
which is a smeared version of the antenna pattern. Using the overlapping MRFs, the rSIR algorithm
reconstructs $T^B$ from the spatially filtered low-resolution sampling provided by the observations. In effect, it
generates an MRF-deconvolved $T^B$ image. Combining multiple orbital passes increases the sampling
density, which improves both the accuracy and resolution of SMAP enhanced-resolution $T^B$ imagery (Long
et al., 2019).
Over Greenland, the rSIR algorithm combines satellite orbital passes that occur between 8 a.m.
and 4 p.m. local time-of-day to reconstruct SMAP enhanced-resolution $T^B$ imagery twice-daily (i.e., morning
and evening orbital pass interval, respectively). $T^B$ imagery is projected on a Northern Hemisphere (NH)
Equal-Area Scalable Earth Grid (EASE-Grid 2.0; Brodzik et al., 2012) at a 3.125 km rSIR grid cell spacing.
The effective resolution for each grid cell is dependent on the number of observations used in the rSIR
reconstruction and is coarser than the rSIR grid cell spacing. While the effective resolution of conventionally
processed SMAP $T^B$ imagery posted on a 25 km grid is ~30 km, the effective resolution of SMAP enhanced-
resolution $T^B$ imagery posted on a 3.125 km grid is ~18 km, an improvement of ~60% (Figs. 1; 2) (Long et
al., 2020).
For our analysis of the percolation facies, we use SMAP enhanced-resolution $T_V^B$ imagery over the
GrIS. Compared to the horizontally-polarized channel, the vertically-polarized channel exhibits decreased
sensitivity to variability in the volumetric fraction of meltwater, which is attributed to reflection coefficient



differences between channels (Miller et al., 2020). Using the vertically polarized channel also results in a
reduced chi-squared error statistic when fitting $T_V^B$ time series to the sigmoid function (see Section 2.4.5).
We construct $T_V^B$ imagery that alternate morning and evening orbital pass observations annually, beginning
and ending just prior to melt onset. The Greenland Ice Mapping Project (GIMP) Land Ice and Ocean
Classification Mask and Digital Elevation Model (Howat et al., 2014) are projected on a NH EASE-Grid 2.0
at a 3.125 km rSIR grid cell spacing. $T_V^B$ imagery between 1 April 2015 and 31 March 2019 are ice sheet-
masked, and an elevation for each rSIR grid cell is calculated.

## 271  2.2    Airborne Ice-Penetrating Radar Surveys

Miller et al., (2020) calibrated the empirical algorithm to map the extent of Greenland's perennial firn
aquifers by correlating a single year of exponentially decreasing temporal L-band signatures (2015-2016)
with coincident perennial firn aquifer detections (2016) identified via MCoRDS. Here, we extend and expand
the calibration of our adapted empirical algorithm to include four years of exponentially decreasing temporal
L-band signatures (2015-2019) correlated with eight years of perennial firn aquifer detections (2010-2017)
and five years of ice slab detections (2010-2014) identified via AR and MCoRDS (Fig. 1c). Our multi-year
calibration technique projects perennial firn aquifer and ice slab detections on three separate NH EASE-
Grids 2.0 at an rSIR grid cell spacing of 3.125 km, consistent with the rSIR grid cell spacing of the SMAP
enhanced-resolution L-band $T_V^B$ imagery. Interannual variability is not resolved in this study, however, it will
be explored further in future work.
An advantage of the multi-year calibration technique as compared to the single-coincident year
calibration technique (Miller et al., 2020) is that it increases the number of rSIR grid cells that can be
assessed. It also provides repeat targets that can account for variability in the dielectric and geophysical
properties that seasonally influence the radiometric temperature and temporal L-band signatures in stable
perennial firn aquifer, ice slab, and perched firn aquifer areas. Uncertainty is introduced by correlating
exponentially decreasing temporal L-band signatures with AR- and MCoRDS-derived detections that are
not coincident in time. The multi-year calibration technique assumes the extent of each area remains stable,
which is not necessarily the case as climate extremes (Cullather et al., 2020) can influence each of these
sub-facies. The assumption of stability neglects boundary transitions in the extent of perennial firn aquifer
areas associated with refreezing of shallow water-saturated firn layers, englacial drainage of meltwater into
crevasses at the periphery (Poinar et al., 2017; Poinar et al, 2019), and transient upslope expansion
(Montgomery et al., 2017). Once formed, ice slabs are essentially permanent features within the upper
snow and firn layers of the percolation facies until they are compressed into glacial ice. However, they may
transition into superimposed ice at the lower boundary of ice slab areas or rapidly expand upslope,
particularly following extreme melting seasons (McFerrin et al., 2019). By our definition, perched firn
aquifers are transient features. Thus, we simply consider our mapped extent a high-probability area for the
preferential formation of each of these sub-facies within the broader percolation facies, with continued
presence dependent on seasonal surface melting and snow accumulation in subsequent years.



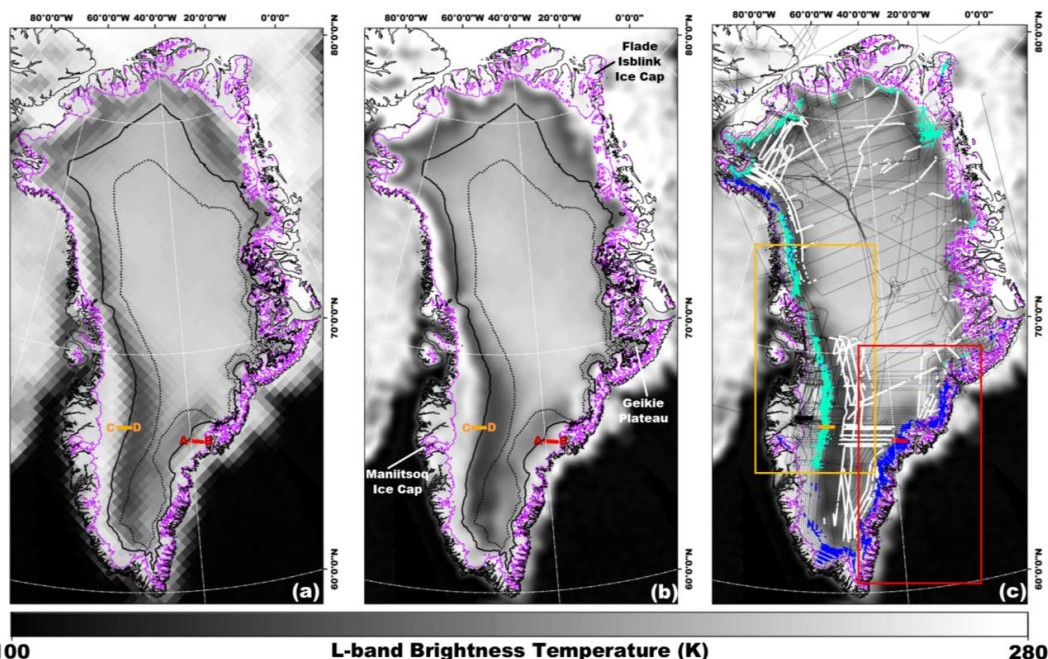

**Figure 1**

*(a) Gridded (25 km), and (b) enhanced-resolution (3.125 km) L-band $T_V^B$ imagery generated using observations collected 15 April 2016 by the microwave radiometer on the SMAP satellite during the evening orbital pass interval over Greenland (Brodzik et al., 2019). The solid black line is the 2000 m.a.s.l. contour, and the black dotted line is the 2500 m.a.s.l. contour (Howat et al., 2014). The purple line is the ice sheet extent (Howat et al., 2014). The black peripheral line is the coast of Greenland and adjacent Ellesmere Island (Wessel and Smith, 1996). The whiter regions of higher $T_V^B$ over the high-elevation (> ~2500 m.a.s.l.) interior are the dry snow facies. The darker grey regions of lower $T_V^B$ are the percolation facies, including ice slabs and perched firn aquifer areas. The whiter regions of higher $T_V^B$ over the coastal areas, peripheral ice caps (e.g., Maniitsoq and Flade Isblink) and nearby islands are perennial firn aquifers, superimposed or glacial ice, land, or spatially integrated L-band emissions. The whiter regions of higher $T_V^B$ outside the ice sheet extent are sea ice. (c) The SMAP enhanced-resolution L-band $T_V^B$ imagery is overlaid with AR- and MCoRDS-derived 2010-2017 perennial firn aquifer (blue shading; Miège et al., 2016), 2010-2014 ice slab (cyan shading; McFerrin et al., 2019), and 2012 spatially coherent melt layer (white shading; Culberg et al., 2021) detections along OIB flight lines (black lines). Overlapping perennial firn aquifer and ice slab detections are interpreted as perched firn aquifer areas. The red and orange boxes in (c) are zoom areas over south eastern Greenland (Fig. 2a), and south western Greenland (Fig. 2b), respectively. The red line is AR radargram profile along perennial firn aquifer transect A-B (Fig. 3a). The orange line is AR radargram profile along ice slab transect C-D (Fig. 3b).*





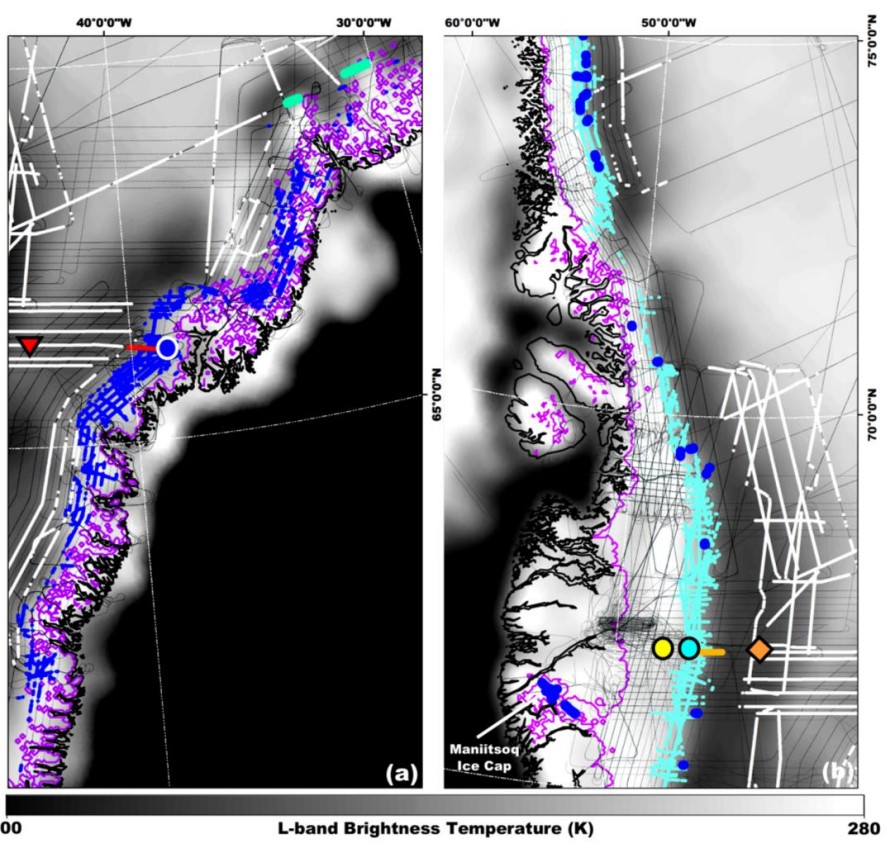

**Figure 2**

*Enhanced-resolution (3.125 km) L-band $T_V^B$ imagery generated using observations collected 15 April 2016*
*by the microwave radiometer on the SMAP satellite during the evening orbital pass interval over (a) south*
*eastern Greenland (Fig. 1c; zoom area in red box), and (b) south western Greenland (Fig. 1c; zoom area*
*in orange box) (Brodzik et al., 2019). The purple line is the ice sheet or ice cap extent (Howat et al., 2014).*
*The black peripheral line is the coast (Wessel and Smith, 1996). (c) The SMAP enhanced-resolution L-*
*band $T_V^B$ imagery is overlaid with AR- and MCoRDS-derived 2010-2017 perennial firn aquifer (blue shading;*
*Miège et al., 2016), 2010-2014 ice slab (cyan shading; McFerrin et al., 2019), and 2012 spatially coherent*
*melt layer (white shading; Culberg et al., 2021) detections along OIB flight lines (black lines). Overlapping*
*perennial firn aquifer and ice slab detections are interpreted as perched firn aquifer areas. The red line is*
*AR radargram profile along perennial firn aquifer transect A-B (Figs. 1; 3a). The orange line is AR radargram*
*profile along ice slab transect C-D (Figs. 1; 3b). The blue circle is a perennial firn aquifer area (Figs. 3a;*
*4a). The cyan circle is a perched firn aquifer area (Figs. 3b; 4b). The orange diamond is a percolation facies*
*area (Fig. 4c). The red triangle is a high-elevation (~2500 m.a.s.l.) percolation facies area (Fig. 4d). The*
*yellow circle is a superimposed ice area (Fig. 4e).*



Annual perennial firn aquifer and ice slab detections that may introduce significant uncertainty into
the multi-year calibration technique include those following the 2010 melting season, which was
exceptionally long (Tedesco et al., 2010), the anomalous 2012 melting season (Nghiem et al., 2012), and
the 2015 melting season which was especially intense in western and northern Greenland (Tedesco et al.,
2016). Following these extreme melting seasons, significant changes in the dielectric and geophysical
properties likely occurred across large portions of the GrIS, including perennial firn aquifer recharging
resulting in increases in meltwater volume and decreases in the depth to the upper surface of stored
meltwater. The formation of expansive near-surface ice slabs (McFerrin et al., 2019) likely resulted in the
formation of more extensive perched firn aquifers during subsequent melting seasons. The upper snow and
firn layers of the dry snow facies and percolation facies were also saturated with relatively large volumetric
fractions of meltwater as compared to the negligible to limited volumetric fractions of meltwater that
percolates during more typical seasonal surface melting on the GrIS. Seasonal meltwater was refrozen into
spatially coherent melt layers following the 2010 and 2012 melting seasons (Culberg et al., 2021) as well
as following the 2015 and 2018 melting seasons (i.e., identified as part of the temporal L-band signature
analysis in this study; see Section 2.4.2).

As compared to ice slabs, which are dense low-permeability solid-ice layers, spatially coherent melt
layers are a network of embedded ice structures primarily consisting of discontinuous horizontally-oriented
ice layers and ice lenses sparsely connected via vertical-oriented ice pipes (Culberg et al., 2021). Ice slabs
are relatively thick (~1 m – 16 m) and form in the high-elevation percolation facies (~2100 m.a.s.l.) at depths
of between ~1 m and 20 m beneath the ice sheet surface following intense seasonal surface melting over
multiple melting seasons (McFerrin et al., 2019). Spatially coherent melt layers are relatively thin (~0.02 cm
- 2 m) and can rapidly form across the entire high-elevation dry snow facies (~3200 m.a.s.l ; Nghiem et al.,
2012) at depths of less than ~1 m beneath the ice sheet surface following a single extreme melting season.
They can further merge together into thicker solid-ice layers following multiple extreme melting seasons
(Culberg et al., 2021). Similar to ice slabs, the formation of spatially coherent melt layers reduces the pore
space within the upper snow and firn layers and may also facilitate lateral meltwater flow with minimum
vertical percolation into the deeper firn layers, thus enhancing meltwater run-off from significantly higher
elevations downslope towards the periphery on accelerated time scales. The formation of spatially coherent
melt layers overlying deeper perennial firn aquifers (e.g., Fig. 3a) will limit or terminate gravity-driven
meltwater drainage and seasonal recharging (Fountain and Walder, 1998), which may eventually
completely refreeze stored meltwater into decimeters thick solid-ice layers overlying deeper glacial ice.
Spatially coherent melt layers are exceptionally bright in AR radargrams (e.g., Fig 3a). The large dielectric
contrast between the spatially coherent melt layer and the overlying, underlying, and interior snow and firn
layers results in high reflectivity at the interfaces. However, electromagnetic energy still propagates
downward through the high reflectivity layer into the deeper firn layers. Culberg et al., 2021) recently
demonstrated mapping the extent of the spatially coherent melt layer formed following the anomalous 2012
melting season (Nghiem et al., 2012) via AR (Figs. 1c; 2).



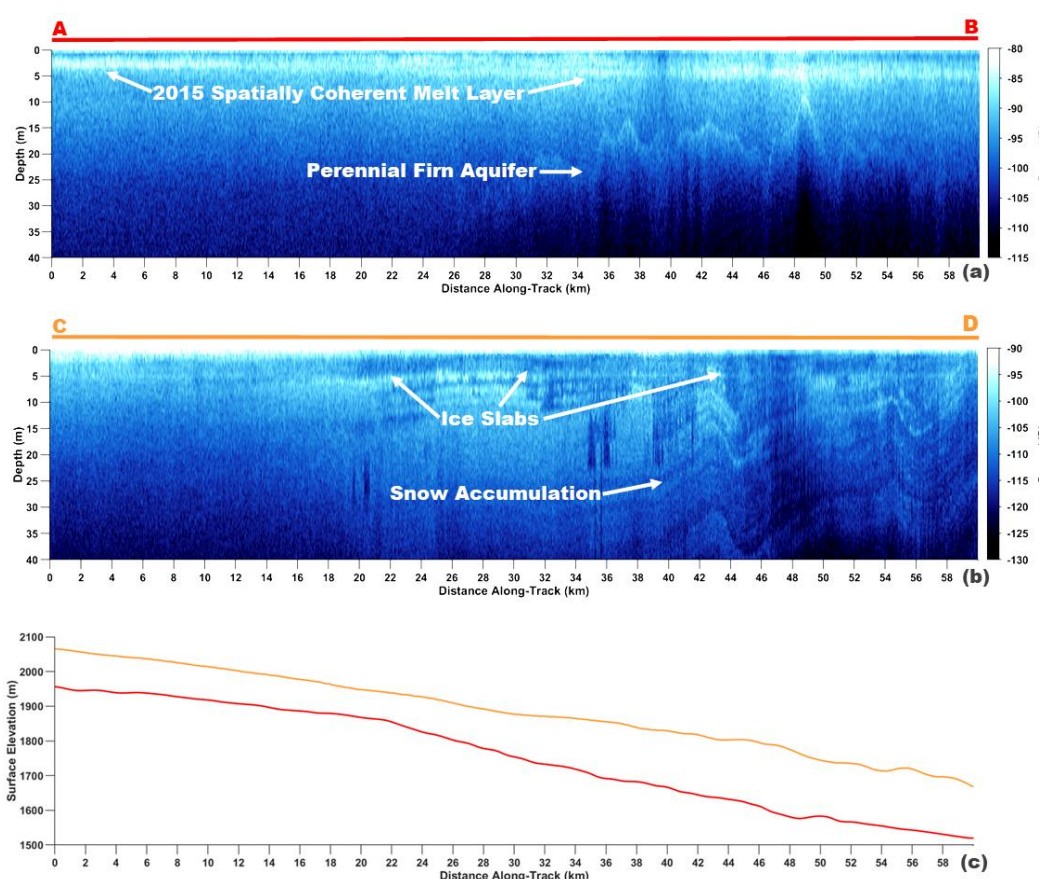


## Figure 3

*AR radargram profile (Rodriguez-Morales et al, 2014) (a) along perennial firn aquifer transect A-B (red line, Figs. 1; 2a) that was collected on 22 April 2017, and (b) ice slab transect C-D (orange line, Figs. 1; 2b) that was collected on 5 May 2017. (c) The corresponding perennial firn aquifer transect A-B elevation profile (red line), and ice slab transect C-D elevation profile (orange line). The exceptionally bright upper surface-parallel reflector in (a) is interpreted as a spatially coherent melt layer that formed following the 2015 melting season. The bright lower reflector in (a) is interpreted as the upper surface of meltwater stored within a perennial firn aquifer. Thick dark surface-parallel regions of low-reflectivity in (b) are interpreted as ice slabs. Alternating sequences of bright and dark surface-parallel reflectors in (b) are interpreted as seasonal snow accumulation layers. A first maximum after maximum gradient re-tracker is used to identify the surface return in each profile. Each profile is flattened so that the depth axis is measured relative to the local elevation. Corresponding elevation profiles in (c) are calculated by subtracting the radar-measured flight clearance over the ice sheet from the aircraft's global positioning system altitude measurements that were coincidently collected along each transect.*



AR and MCoRDS (Rodriguez-Morales et al, 2014) were flown over the GrIS on a P-3 aircraft in
April and May between 2010 and 2017. The AR instrument operates at a center frequency of 750 MHz with
a bandwidth of 300 MHz, resulting in a range resolution in firn of 0.53 m (Lewis et al., 2015). The collected
data have an along-track resolution of ~30 m with 15 m spacing between traces in the final processed
radargrams. At a nominal flight altitude of 500 m above the ice sheet surface, the cross-track resolution
varies between 20 m for a smooth surface, to 54 m for a rough surface with no appreciable layover. The
MCoRDS instrument operated at three different frequency configurations: (1) a center frequency of 195
MHz with a bandwidth of 30 MHz (2010-2014, 2017, 2018), (2) a center frequency of 315 MHz with a band
width of 270 MHz (2015), and (3) a center frequency of 300 MHz with a bandwidth of 300 MHz (2016). The
vertical range resolution in firn for each of these frequency configurations is 5.3 m, 0.59 m, and 0.53m,
respectively (CReSIS, 2016). The collected data have an along-track resolution of ~25 m with 14 m spacing
between traces in the final processed radargrams. At the same nominal flight altitude of 500 m, the cross-
track resolution varies between 40 m for a smooth surface in the highest bandwidth configuration, to 175
m for a rough surface with no appreciable layover in the lowest bandwidth configuration.

The multi-year calibration technique uses perennial firn aquifer detections previously identified
along OIB flight lines via AR (2010-2014) and MCoRDS (2015-2017) radargram profiles and the
methodology described in Miège et al. (2016). Bright lower reflectors that undulate with the local
topographic gradient underneath which reflectors are absent in the percolation facies are interpreted as the
upper surface of meltwater stored within perennial firn aquifers (e.g., Fig. 3a). The large dielectric contrast
between refrozen and water-saturated firn layers results in high reflectivity at the interface. However, the
presence of meltwater increases attenuation, limiting the downward propagation of electromagnetic energy
through the water-saturated firn layer. The total number of AR derived perennial firn aquifer detections is
~325,000, corresponding to a total extent of ~98 $km^2$. The analysis assumes a smooth surface, which is
typical of much of the percolation facies, and a grid cell size of 15 m x 20 m. The total number of MCoRDS-
derived perennial firn aquifer detections is ~142,000, corresponding to a total extent of ~80 $km^2$. This also
assumes a smooth surface, and a grid cell size of 14 m x 40 m. The combined total number of grid cells
(~467,000) and total extent (~178 $km^2$) is significantly larger than the total number of MCoRDS-derived grid
cells (~78,000) and total extent (~44 $km^2$) calculated for 2016 (Miller et al., 2020). Perennial firn aquifer
detections are mapped in western, southern, and south and central eastern Greenland as well as the
Maniitsoq and Flade Isblink Ice Caps (Figs. 1c; 2a). We project AR- and MCoRDS-derived perennial firn
aquifer detections on the NH EASE-Grid 2.0 at an rSIR grid cell spacing of 3.125 km. Each rSIR grid cell
has an extent of ~10 $km^2$. The total number of rSIR grid cells with at least one perennial firn aquifer detection
is ~800, corresponding to a total extent of ~8000 $km^2$. However, given the limited AR and MCoRDS grid
cell coverage, less than ~1% of the rSIR grid cell extent has radargram coverage. As compared to the total
number of MCoRDS-derived perennial firn aquifer detections (~780) calculated for 2016 (Miller et al., 2020),
the total number of rSIR grid cells with at least one detection is only increased by ~20 for the multi-year
calibration technique, corresponding to an increased total extent of ~200 $km^2$.





We also use ice slab detections previously identified along OIB flight lines via AR (2010-2014)
radargram profiles and the methodology described in McFerrin et al. (2019) in the multi-year calibration
technique. Thick dark surface-parallel regions of low-reflectivity in the percolation facies are interpreted as
ice slabs (Fig. 3b). The large dielectric contrast between ice slabs and the overlying and underlying snow
and firn layers results in high reflectivity at the interfaces. However, electromagnetic energy is not scattered
or absorbed within the homogeneous ice slab, it instead propagates downward through the layer and into
the deeper firn layers. The total number of AR-derived ice slab detections is ~505,000, corresponding to a
total extent of ~283 km². Ice slab detections are mapped in western, central and north eastern, and northern
Greenland as well as the Flade Isblink Ice Cap (Figs. 1c; 2b). We project the AR-derived ice slab detections
on the NH EASE-Grid 2.0 at an rSIR grid cell spacing of 3.125 km. The total number of rSIR grid cells with
at least one ice slab detection is ~2000, corresponding to a total extent of ~20,000 km².However, less than
~2% of the rSIR grid cell extent has radargram coverage.
We detect perched firn aquifer areas by comparing the AR- and MCoRDS-derived perennial firn
aquifer and ice slab detections projected on the NH EASE-Grid 2.0 and then identify overlapping rSIR grid
cells. The total number of AR-derived perched firn aquifer detections is ~75,000, corresponding to a total
extent of ~23 km². The total number of MCoRDS-derived perched firn aquifer detections is ~20,
corresponding to a near-negligible extent (~0.006 km²). Perched firn aquifer detections are mapped in
western, and central eastern Greenland as well as the Flade Isblink Ice Cap (Figs. 1c; 2b).
The total number of rSIR grid cells with at least one perched firn aquifer detection is ~200,
corresponding to a total extent of ~2000 km². However, similar to the other sub-facies, less than ~1% of
the rSIR grid cell extent has radargram coverage. The total number of AR- and MCoRDS-derived perennial
firn aquifer, ice slab, and perched firn aquifer detections that we project on three separate NH EASE-Grids
2.0, the associated total number of rSIR grid cells that we use in the calibration of our adapted empirical
algorithm, and the coverage of detections and rSIR grid cells over each of the three sub-facies within the
broader percolation facies are summarized in Table 1.

**Table 1.** *The total number of airborne ice penetrating radar survey detections (2010-2017), the associated*
*total number of rSIR grid cells, and the coverage of detections and rSIR grid cells over perennial firn aquifer,*
*ice slab, and perched firn aquifer areas.*

|  | Detections | Coverage (km²) | rSIR Grid Cells | Coverage (km²) |
|---|---|---|---|---|
| **Perennial Firn Aquifers** | ~467,000 | ~178 | ~80 | ~8000 |
| **Ice Slabs** | ~505,000 | ~283 | ~2000 | ~20,000 |
| **Perched Firn Aquifers** | ~75,000 | ~23 | ~200 | ~2000 |





## 2.4  Empirical Algorithm

### 2.4.1  Greenland's Ice Facies

Greenland's ice facies (i.e., dry snow facies - percolation facies - wet snow facies - ablation facies) were first described in detail by Benson et al., (1960), and were shown to represent the GrIS's response to climate. Evolution of the boundaries of Greenland's ice facies are often used as an indicator of climate change. Early studies using field-based (Jezek et al., 1994; Zabel et al., 1995), airborne (Swift et al., 1985; Bindschadler et al., 1987; Rignot et al., 1993; Jezek et al., 1993), and satellite (Fahnestock et al., 1993; Long and Drinkwater, 1994; Parrington, 1998) synthetic aperture radars and radar scatterometers operating at frequencies between Ku-band (13 GHz) and P-band (0.4 GHz) have demonstrated the exceptional capabilities of microwave sensors for mapping Greenland's ice facies. Early airborne studies using C-band microwave radiometry (Swift et al., 1985), and more recent studies using L-band microwave radiometry (Jezek et al. 2018) have demonstrated similar capabilities. In this study, we extend these capabilities to include satellite L-band microwave radiometry. We delineate the boundaries of the percolation facies relative to the adjacent dry snow facies (i.e., where negligible seasonal surface melting occurs) and wet snow facies (i.e., where snow layers are fully water-saturated during the melting season and subsequently refreeze as superimposed ice overlying deeper glacial ice). And, we further identify sub-facies (i.e., perennial firn aquifer, ice slabs, and perched firn aquifers) within the broader percolation facies that are currently experiencing rapid expansion (McFerrin et al., 2019; Culberg et al., 2021) as Greenland's climate continues to warm (Hanna et al., 2013; Cullather et al., 2020) and seasonal surface melting increases in extent, intensity, and duration (Steffen et al., 2004; Tedesco e al., 2008; Tedesco et al., 2011; Nghiem et al., 2012; Tedesco et al., 2016; Tedesco and Fettweis, 2020; Tedesco and Fettweis, 2020). Higher frequency microwave sensors provide shallower penetration depths, and an increased sensitivity to snow grain size, layering, embedded ice structures (Long and Drinkwater, 1994; Drinkwater et al., 2001) and stored meltwater (Jezek et al., 1993; Miller, 2019) within the upper snow and firn layers of the percolation facies. Lower frequencies provide deeper penetration depths and a range of sensitivities to embedded ice structures (Jezek et al., 1993; Jezek et al., 2018) and stored meltwater (Miller et al., 2020) in the deeper firn layers.

### 2.4.2  Temporal L-band signatures over the percolation facies

Microwave brightness temperature ($T^B$) expresses the satellite-observed magnitude of thermal emission and is influenced by the observation geometry as well as the dielectric and geophysical properties of the ice sheet (Ulaby et al., 2014). The most significant geophysical property influencing $T^B$ is the volumetric fraction of meltwater within the snow and firn pore space (Mätzler and Hüppi, 1989). During the melting season, the upper snow and firn layers of the percolation facies are saturated with large volumetric fractions of meltwater that percolates vertically into the deeper firn layers (Benson, 1960; Humphrey et al., 2012). Increases in the volumetric fraction of meltwater results in rapid relative increases in the imaginary part of





the complex dielectric constant (Tiuiri et al., 1984), with corresponding increases in $T^B$. This increase is
attributed to a decrease in volume scattering, and penetration depth. The L-band penetration depth can
rapidly decrease from ~tens to hundreds of meters, to less than ~a meter, dependent on the local snow
and firn conditions, and englacial firn hydrological features. Surface and subsurface water-saturated snow
and firn layers and embedded ice structures subsequently refreeze. During the freezing season, decreases
in the volumetric fraction of meltwater results in rapid relative decreases in the imaginary part of the complex
dielectric constant, with corresponding decreases in $T^B$. This increase is attributed to an increase in volume
scattering, and penetration depth. The L-band penetration depth increases back to ~tens to hundreds of
meters on variable time scales.
We analyze melting and freezing seasons in temporal L-band signatures exhibited in $T_V^B$ time series
(1 April 2015 - 31 March 2019) over and near AR- and MCoRDS-derived perennial firn aquifer, ice slab,
and perched firn aquifer detections projected on NH EASE-Grids 2.0 (Fig. 4). We project ice surface
temperature data calculated using thermal infrared brightness temperature collected by the Moderate
Resolution Imaging Spectroradiometer (MODIS) on the Terra and Aqua satellites (i.e., Hall et al., 2012) on
coincident NH EASE-Grids 2.0 at a 3.125 km rSIR grid cell spacing. We then derive melt onset and surface
freeze-up dates (2015-2019) for each rSIR grid cell using the methodology described in Miller et al., (2020).
We set a threshold of ice surface temperature >−1°C for meltwater detection (Nghiem et al., 2012),
consistent with the ±1°C accuracy of the ice surface temperature data. For temperatures that are close to
0°C, ice surface temperatures are closely compatible with contemporaneous NOAA near-surface air
temperature data (Shuman et al., 2014). Melt onset and surface freeze-up dates are overlaid on $T_V^B$ time
series to partition the melting and freezing seasons. Melt onset dates occur between ~April and July, and
surface freeze-up dates occur between ~July and September. The melting season increases in duration
moving downslope from the dry snow facies, and ranges from a single day in the highest elevations (>2500
m) of the percolation facies, to ~150 days in the ablation facies. Similarly, the associated freezing season
decreases in duration moving downslope and ranges from between ~215 days and 365 days.
Over perennial firn aquifer areas (e.g., Figs. 1c; 2a; 4a), $T_V^B$ is radiometrically warm during the
melting season. Vertically percolating meltwater and gravity-driven meltwater drainage seasonally
recharges perennial aquifers at depth (Fountain and Walder et al., 1998). Maximum values range from
between ~200 K and 275 K during seasonal surface melting. Temporal L-band signatures exhibit increases
on time scales of ~days to weeks following the melt onset date, and melting seasons range from between
~75 and 100 days. $T_V^B$ remains radiometrically warm during the freezing season as a result of latent heat
continuously released by the slow refreezing of the deeper firn layers that are saturated with large
volumetric fractions of meltwater (Miller et al, 2020). Minimum values range from between ~180 K and 250
K following the surface freeze-up date. L-band emissions from the radiometrically warm upper snow and
firn layers decrease during the freezing season as embedded ice structures slowly refreeze at increased
depths below the ice sheet surface (Miller et al., 2020). Temporal L-band signatures exhibit exponential
decreases on time scales of ~months that approach and sometimes achieve relatively stable $T_V^B$ values,



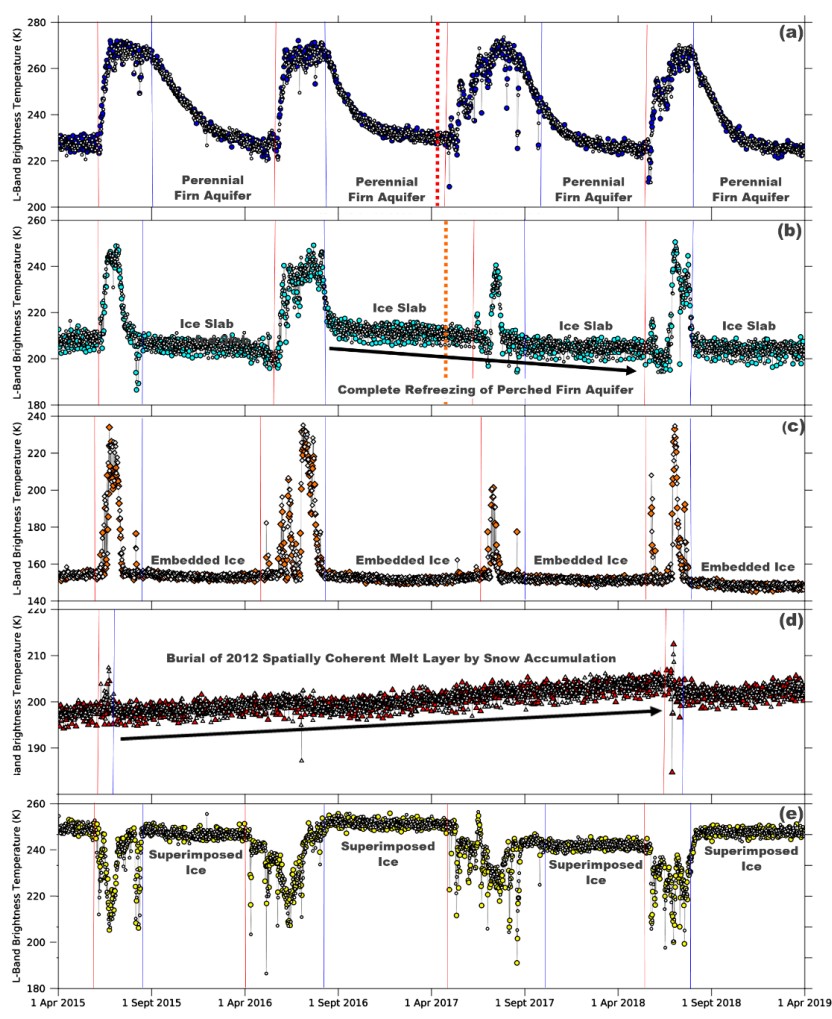


## Figure 4

*Temporal L-band signatures that alternate morning (white symbols) and evening (colored symbols) orbital pass interval enhanced-resolution $T_V^B$ generated using observations collected over the GrIS by the microwave radiometer on the SMAP satellite (Brodzik et al., 2019) over (a) perennial firn aquifer area (blue circles; Figs. 2a; 3a), (b) perched firn aquifer area (cyan circles; Figs. 2b; 3b), (c) percolation facies area (orange diamonds; Fig 2b), (d) high-elevation (~2500 m.a.s.l.) spatially coherent melt layer area (red triangles; Fig. 2a), and (e) superimposed ice area (yellow circles; Fig. 2b). Melt onset (red lines) and surface freeze-up (blue lines) dates are derived from thermal infrared $T^B$ collected by MODIS on the Terra and Aqua satellites (Hall et al, 2012). AR radargram profile along perennial firn aquifer transect A-B (red dashed line; Figs. 1; 2a; 3a) that was collected on 22 April 2017, and ice slab transect C-D (orange dashed line; Figs. 1; 2b; 3b) that was collected on 5 May 2017.*





and freezing seasons range from between ~265-290 days. $T_V^B$ often decreases by more than ~50 K during
the freezing season (e.g., Fig. 4a), representing the descent of the upper surface of stored meltwater by
~tens of meters (Miège et al., 2016).

Over ice slab and perched firn aquifer areas (e.g., Figs. 1c; 2b; 5b), $T_V^B$ is typically radiometrically

colder than over perennial firn aquifer areas during the melting season. The presence of dense low-
permeability solid-ice layers (e.g., Fig. 3b) reduces the snow and firn pore space available to store seasonal
meltwater at depth. Meltwater may alternatively run-off downslope towards the wet snow facies. Maximum
values range from between ~170 K and 260 K during seasonal surface melting. Temporal L-band signatures
exhibit increases on time scales of ~days to weeks following the melt onset date, and melting seasons
range from between ~60 and 90 days. $T_V^B$ is also typically radiometrically colder than over perennial firn
aquifer areas during the freezing season as a result of the absence of meltwater stored at depth (i.e. ice
slab areas), or the presence of limited volumetric fractions of meltwater stored at depth in shallow water-
saturated firn layers (i.e. perched firn aquifer areas). Minimum values range from between ~130 K and 240
K following the surface freeze-up date. Temporal L-band signatures exhibit exponential decreases on time
scales of ~weeks to months that often achieve relatively stable $T_V^B$ values, and freezing seasons range from
between ~275-305 days. Exponentially decreasing temporal L-band signatures sometimes transition to
linearly decreasing on time scales of ~years following the surface freeze-up date (e.g., between
~September 2016 and May 2018 in Fig. 4b). We infer this indicates the formation and subsequent refreezing
of a shallow perched firn aquifer on top of a buried ice slab or other semi-impermeable layer. As compared
to the large $T_V^B$ decreases in percolation facies areas, $T_V^B$ decreases over perched firn aquifer areas are as
small as ~a few K annually, which represents the descent of the upper surface of stored meltwater by
~meters rather than by ~tens of meters.

Over other percolation facies areas, where seasonal meltwater is fully refrozen and stored

exclusively as embedded ice (e.g., Fig. 4c), $T_V^B$ is typically radiometrically colder than over perennial firn
aquifer, ice slab, and perched firn aquifer areas during the melting season. Maximum values range from
between ~150 K and 200 K during seasonal surface melting. Temporal L-band signatures exhibit increases
on time scales of ~days to weeks following the melt onset date, and melting seasons range from between
~1 and 60 days. $T_V^B$ is also typically radiometrically cold during the freezing season. Minimum values range
from between ~130 K and 180 K following the surface freeze-up date. Temporal L-band signatures exhibit
exponential decreases on time scales of ~days to weeks and achieve relatively stable $T_V^B$ values, and
freezing seasons range from between ~305-364 days. However, over the highest elevations (> ~2500
m.a.s.l.) of the percolation facies approaching the dry snow line, where seasonal surface melting and the
formation of embedded ice structures is limited, $T_V^B$ remains radiometrically warm during the freezing
season. Minimum values range from between ~180 K and 220 K following the surface freeze-up date. We
infer $T_V^B$ decreases, sometimes step-responses exceeding ~10 K, that follow the surface freeze-up date
(e.g., between April 2018 and September 2018 in Fig. 4c) are a result of an increase in volume scattering
from newly formed embedded ice structures within a spatially coherent melt layer. We also infer that





temporal L-band signatures that increase several K on time scales of ~years (e.g., between ~April 2015
and April 2018 in Fig. 4c) indicate the burial of spatially coherent melt layers formed following the 2010,
2012, 2015, and 2018 melting seasons by snow accumulation.

Exponentially decreasing temporal L-band signatures transition smoothly between perennial firn
aquifer, ice slab, perched firn aquifer, and other percolation facies areas – there are no distinct temporal L-
band signatures that delineate boundaries between these sub-facies. Boundary transitions between other
facies, however, are delineated both above and below the percolation facies. Over the dry snow facies
(e.g., Fig. 4d), $T_V^B$ is radiometrically warm during the melting and freezing seasons. Values range from
between ~200 K and 240 K. While $T_V^B$ is known to be relatively stable in the dry snow facies, temporal L-
band signatures that increase on time scales of ~years are observed throughout this region at elevations
as high as Summit Station (~3200 m.a.s.l), similar to those observed in the highest elevations (> ~2500
m.a.s.l.) of the percolation facies. We infer increasing temporal L-band signatures indicate the burial of the
spatially coherent melt layer formed following the anomalous 2012 melting season (Nghiem et al., 2012) by
snow accumulation (Culberg et al., 2021). Over the wet snow facies (e.g., Fig. 4e), where seasonal
meltwater is fully refrozen and stored as superimposed ice, $T_V^B$ is radiometrically warm during the melting
season. Maximum values range from between ~230 K and 250 K during seasonal surface melting. As
compared to the percolation facies, where temporal L-band signatures exhibit rapid increases following
melt onset, temporal L-band signatures reverse and exhibit decreases on time scales of ~days to weeks,
and melting seasons that range between ~90-120 days. We infer these reversals are the result of high
reflectivity and attenuation at the fully water-saturated snow layer and/or at the wet, rough superimposed
ice-air interface. Meltwater runs-off superimposed ice downslope towards the ablation facies in the wet
snow facies. $T_V^B$ remains radiometrically warm during the freezing season. Minimum values range from
between ~230 K and 250 K following seasonal surface melting. Temporal L-band signatures exhibit
increases on time scales of ~days that achieve relatively stable $T_V^B$ values, and freezing seasons range
from between ~245 and 275 days.

The MODIS-derived total number of days in the melting and freezing seasons estimated from melt
onset and surface freeze-up dates, the SMAP-derived maximum and minimum vertically-polarized L-band
brightness temperature, and the time scales of exponential decrease following the surface freeze-up date
estimated for each $T_V^B$ time series for rSIR grid cells over perennial firn aquifer, ice slab, perched firn aquifer,
and other percolation facies areas as well as for the dry snow facies, and the wet snow facies are
summarized in Table 2.





**Table 2.** *The MODIS-derived total number of days in the melting and freezing seasons (2015-2019), the*
*SMAP-derived maximum vertically-polarized L-band brightness temperature ($T^B_{V,max}$), the minimum*
*vertically-polarized L-band brightness temperature ($T^B_{V,min}$), and the time scale scales of exponential*
*decrease following the surface freeze-up date (1 April 2015 - 31 March 2019) for perennial firn aquifer, ice*
*slab, perched firn aquifer, and other percolation facies areas as well as for the dry snow facies and the wet*
*snow facies.*

|  | Melting Season (days) | Freezing Season (days) | $T^B_{V,max}$ (K) | $T^B_{V,min}$ (K) | Exponential Decrease (time scale) |
|---|---|---|---|---|---|
| **Perennial Firn Aquifers** | ~75 - 100 | ~265 - 290 | ~200 - 275 | ~180 – 250 | ~weeks – months |
| **Ice Slabs / Perched Firn Aquifers** | ~60 -90 | ~275 - 305 | ~170 - 260 | ~130 – 240 | ~days - Weeks |
| **Percolation Facies** | ~1 - 60 | ~305 - 364 | ~150 - 200 | ~130 – 220 | ~days |
| **Dry Snow Facies** | - | 365 | ~200 - 240 | ~200 – 240 | - |
| **Wet Snow Facies** | ~90 - 120 | ~245 - 275 | ~230 - 250 | ~230 – 250 | - |


### 623     2.4.3   L-band geophysical-brightness temperature model

Based on our analysis of $T^B_{V,max}$ and $T^B_{V,min}$ values in temporal L-band signatures over the percolation facies,
we derive a 'firn saturation' parameter using the simple two-layer L-band geophysical-brightness
temperature model described in Ashcraft and Long (2006). The firn saturation parameter is similar to the
'melt intensity' parameter derived in Hicks and Long (2011) that uses enhanced resolution vertically-
polarized Ku-band radar backscatter  imagery (2003) collected by the SeaWinds radar scatterometer that
was flown in tandem on NASA's Quick SCATterometer (QuikSCAT) satellite (Tsai et al., 2000) and JAXA's
Advanced Earth Observing Satellite 2 (ADEOS-II) (Freilich et al., 1994). We use the firn saturation
parameter to estimate the maximum seasonal volumetric fraction of meltwater within the saturated upper
snow and firn layers of the percolation facies using $T^B_{V,max}$ and $T^B_{V,min}$ values extracted from $T^B_V$ time series
(1 April 2015 - 31 March 2019). We calculate the firn saturation parameter for each rSIR grid cell within the
ice sheet-masked extent of the GrIS as part of our adapted empirical algorithm (see Section 2.4.5).

We first describe the geophysical model as follows. We assume a base layer underlying a water-

saturated firn layer with a given depth and volumetric fraction of meltwater. Each of the layers is
homogenous. We next describe $T^B_V$ from the geophysical model (Eq. 1). The ice sheet is discretely layered
(i.e., two-layers; the base layer, and the water-saturated firn layer) to calculate $T^B_V$ at an oblique incidence
angle. Emissions from the base layer are a function of both the macroscopic roughness and the dielectric
properties of the layer. They occur in conjunction with volume scattering at depth, and are locally dependent





on englacial firn hydrological features, including embedded ice structures, spatially coherent melt layers,
ice slabs, and perennial and perched firn aquifers. Reflectivity at depth (i.e., at the base layer-water-
saturated firn layer interface), and at the ice sheet surface (i.e., at the water-saturated firn layer-air interface)
is neglected. The contribution from each layer is individually calculated.
The two-layer L-band geophysical-brightness temperature model is represented analytically by
$$T_{V,max}^B = T\left(1 - e^{-\kappa_e d sec\theta}\right) + T_{V,min}^B e^{-\kappa_e d sec\theta} \, ,$$   (Eq. 1)
where $T_{V,max}^B$ is the maximum vertically-polarized L-band brightness temperature at the ice sheet
surface, $T_{V,min}^B$ is the minimum vertically-polarized L-band brightness temperature emitted from the base
layer, $T$ is the physical temperature of the water-saturated firn layer, $\theta$ is the transmission angle, $\kappa_e$ is the
extinction coefficient, and $d$ is depth.
We invert Eq. 1 and solve for the firn saturation parameter ($\xi$)
$$\xi = ln\left(\frac{T_{V,max}^B - T}{T_{V,min}^B - T}\right) cos\theta \, ,$$   (Eq. 2)
where $\xi = \kappa_e d$. The maximum vertically-polarized L-band brightness temperature asymptotically approaches
the physical temperature of the water-saturated firn layer as the extinction coefficient and the depth of the
water-saturated firn layer increases. The extinction coefficient is defined as the sum of the Raleigh
scattering coefficient ($\kappa_s$) and the absorption coefficient ($\kappa_a$). For water-saturated firn, absorption dominates
over scattering, and increases in the extinction coefficient are controlled by the volumetric fraction of
meltwater ($m_v$). We assume that thicker water-saturated firn layers with larger volumetric fractions of
meltwater generate higher firn saturation parameter values. However, the thickness of the water-saturated
firn layer is limited by the L-band penetration depth. Theoretical L-band penetration depths calculated for a
water-saturated firn layer range from between ~10 m for small volumetric fractions of meltwater ($m_v$<1%),
and ~1 cm for large volumetric fractions of meltwater ($m_v$=20%) (Fig. 5). Large volumetric fractions of
meltwater results in high reflectivity and attenuation at the interface between water-saturated firn layers and
the overlying refrozen firn layers, and between glacial ice or a semi-impermeable layer and the overlying
water-saturated firn layers, and a radiometrically cold firn layer (e.g., Fig 5e).

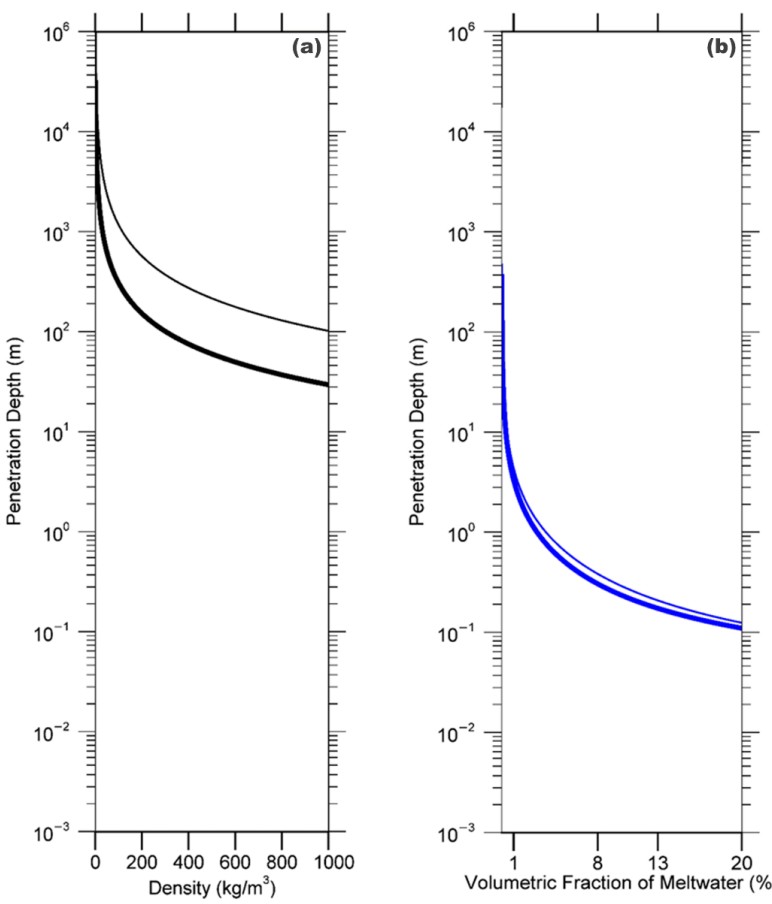


**Figure 5**

*Theoretical L-band penetration depths for (a) refrozen, and (b) water-saturated firn. Penetration depths*

$\left( \frac{1}{\kappa_s + \kappa_a} \right)$ *are calculated as a function of the Raleigh scattering coefficient ($\kappa_s$; Eq. 8) and the absorption*

*coefficient ($\kappa_a$; Eq. 10), which are functions of the dielectric and geophysical properties of the GrIS. The*

*complex dielectric constant is calculated using the empirically derived models described in Tiuri et al.,*

*(1984). Refrozen firn penetration depths are calculated as a function of firn density ($\rho_{firn}$), and the curves*

*are plotted for snow grain radii ($r$) set to $r$ =0.5 mm (upper curve), and $r$ =4 mm (lower curve). Water-*

*saturated firn penetration depths are calculated as a function of the volumetric fraction of meltwater ($m_v$),*

*and the curves are plotted for firn density set to $\rho_{firn}$=400 kg/m³ (upper curve), and $\rho_{firn}$=917 kg/m³ (lower*

*curve). Given the complexity of modeling embedded ice structures, they are excluded from the penetration*

*depth calculation. Increases in the volumetric fraction of embedded ice in the firn will result in an increase*

*in volume scattering, which will decrease and compress the distance between the penetration depth curves*

*for both refrozen and water-saturated firn.*





### 2.4.4 Continuous logistic model

We adapt our previously developed empirical algorithm to map the extent of Greenland's perennial firn aquifers (Miller et al., 2020) to also map the extent of ice slab and perched firn aquifer areas. The empirical algorithm is derived from the continuous logistic model, which is based on a differential equation that models the decrease in physical systems as a function of time using a set of sigmoidal curves. These curves begin at a maximum value with an initial interval of decrease that is approximately exponential. Then, as the function approaches its minimum value, the decrease slows to approximately linear. Finally, as the function asymptotically reaches its minimum value, the decrease exponentially tails off and achieves stable values. We use the continuous logistic model to parametrize the refreezing rate within the saturated upper snow and firn layers of the percolation facies using $T_V^B$ time series (1 April 2015 - 31 March 2019) that are partitioned using $T_{V,max}^B$ and $T_{V,min}^B$ values. We calculate the refreezing rate for each rSIR grid cell within the percolation facies extent as part of our adapted empirical algorithm (see Section 2.4.5).

The continuous logistic model is described by a differential equation known as the logistic equation

$$\frac{dx}{dt} = \zeta x(1-x) \qquad \text{(Eq. 3)}$$

that has the solution

$$x(t) = \frac{1}{1+\left(\frac{1}{x_o}-1\right)e^{-\zeta t}}, \qquad \text{(Eq. 4)}$$

where $x_o$ is the function's initial value, $\zeta$ is the function's exponential rate of decrease, and $t$ is time. The function $x(t)$ is also known as the sigmoid function. We use the sigmoid function to model the exponentially decreasing temporal L-band signatures observed over the percolation facies as a set of decreasing sigmoidal curves.

We first normalize $T_V^B$ time series for each rSIR grid cell

$$T_{V,N}^B(t) = \frac{T_V^B(t)-T_{V,min}^B}{T_{V,max}^B-T_{V,min}^B}, \qquad \text{(Eq. 5)}$$

where $T_{V,min}^B$ is the minimum vertically-polarized L-band brightness temperature, and $T_{V,max}^B$ is the maximum vertically-polarized L-band brightness temperature. We then apply the sigmoid fit

$$T_{V,N}^B\left(t \in [t_{max}, t_{min}]\right) = \frac{1}{1+\left(\frac{1}{T_{V,N}^B(tmax)}-1\right)e^{-\zeta t}}. \qquad \text{(Eq. 6)}$$



$T_{V,N}^B \left( t \in [t_{max}, t_{min}] \right)$ is the normalized vertically-polarized L-band brightness temperature on the time
interval $t \in [t_{max}, t_{min}]$, where $t_{max}$ is the time the function achieves a maximum value, and $t_{min}$ is the
time the function achieves a minimum value. The initial normalized vertically-polarized L-band brightness
temperature ($T_{V,N}^B(t_{max})$) is the function's maximum value. The final normalized vertically-polarized L-band
brightness temperature ($T_{V,N}^B(t_{min})$) is the function's minimum value. The function's exponential rate of
decrease represents the refreezing rate parameter ($\zeta$). An example set of simulated sigmoidal curves is
shown in Fig. 6.

**2.4.5   SMAP-derived perennial firn aquifer, ice slab, and perched firn aquifer maps**
Our adapted empirical algorithm uses ice sheet-masked SMAP enhanced-resolution $T_V^B$ imagery over the
GrIS that alternates morning and evening orbital pass observations annually, beginning and ending just
prior to melt onset. Our algorithm is implemented in two steps: (1) mapping the extent of the percolation
facies using the firn saturation parameter derived from the L-band geophysical-brightness temperature
model (see Section 2.4.3), and (2) mapping the extent of perennial firn aquifer, ice slab, and perched firn
aquifer areas over the percolation facies using the continuous logistic model (see Section 2.4.4) we
calibrate using airborne ice-penetrating radar detections projected on three separate NH EASE-Grids 2.0
(see Section 2.2).
Using Eq. 2, we first set a threshold for the firn saturation parameter ($\xi_T$) defined by the relationship

$$\xi_T = (\kappa_s + \kappa_a)d \leq \xi \ . \tag{Eq. 7}$$

We calculate the Raleigh scattering coefficient ($\kappa_s$) in Eq. 7 using

$$\kappa_s = N_d \frac{8}{3} k_o^4 r^6 \left| \frac{\varepsilon_r - 1}{\varepsilon_r + 2} \right|^2 , \tag{Eq. 8}$$

where $N_d$ is the particle density, $k_o$ is the wave number of the background medium of air, $r$ is the snow
grain radius set to $r$=2 mm, and $\varepsilon_r$ is the complex dielectric constant. The particle density is defined by

$$N_d = \frac{\rho_{firn}}{\rho_{ice}} \frac{1}{\frac{4}{3}\pi r^3} , \tag{Eq. 9}$$

where $\rho_{firn}$ is firn density, which we set to $\rho_{firn}$=400 kg/m³, and $\rho_{ice}$ is ice density, which we set to $\rho_{ice}$=917
kg/m³. Our grain radius and firn density estimates are consistent with measurements within the upper snow
and firn layers of the percolation facies of south eastern Greenland at the Helheim Glacier field site (Fig.


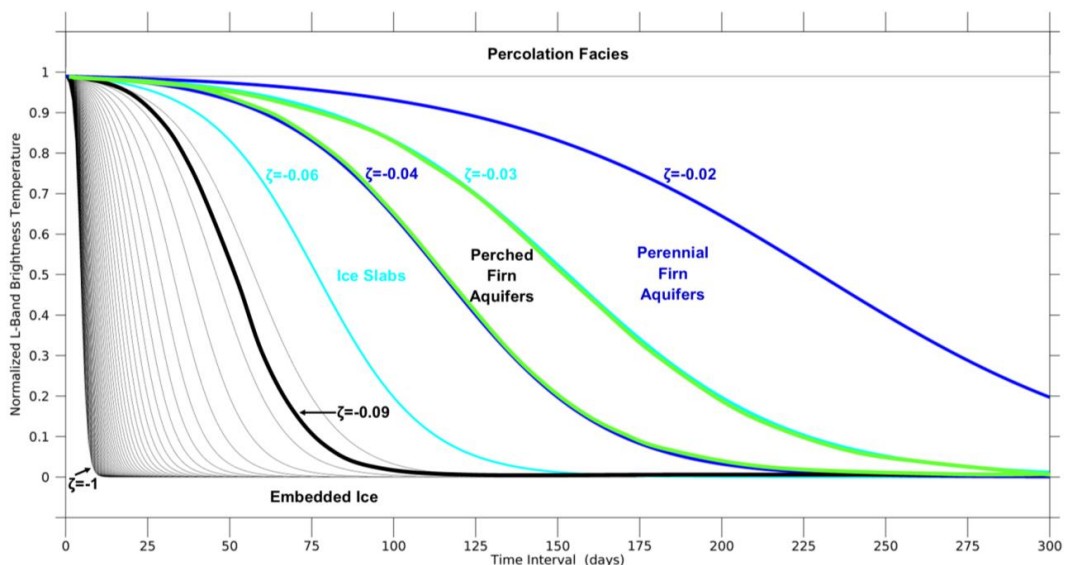


### Figure 6

*Example set of simulated sigmoidal curves that represent our model of the exponentially decreasing*

*temporal L-band signatures predicted over the percolation facies. The initial normalized vertically-polarized*

*L-band brightness temperature was fixed at a value of $T_{V,N}^B(t_{max}) = 0.99$, and the time interval was set to a*

*value of $t \in [\,t_{max}\,,\,t_{min}] = 300$ observations. The refreezing rate parameter was set to values between $\zeta =$*

*$[\,-1,\,0\,]$ incremented by steps of 0.02. The blue lines correspond to the interval $\zeta \in [\,-0.04,\,-0.02\,]$ and*

*produce curves similar to those observed over perennial firn aquifer areas. The cyan lines correspond to*

*the interval $\zeta \in [-0.06,\,-0.03\,]$ and produce curves similar to those observed over ice slab areas. The green*

*lines correspond to the interval $\zeta \in [\,-0.04\,,-0.03\,]$ and produce curves similar to those observed over*

*perched firn aquifer areas. The black line is the observed lower bound ($\zeta = -0.09$) of the refreezing rate*

*parameter of partitioned $T_V^B$ time series (1 April 2015 - 31 March 2019) iteratively fit to the sigmoid function*

*(see Section 3).*




2a; blue circle; Fig. 4a), where in situ perennial firn aquifer measurements have recently been collected
(Miller et al., 2017).
We calculate the absorption coefficient ($\kappa_a$) in Eq. 7 using

$$\kappa_a = -2k_o \mathfrak{I}\{\sqrt{\varepsilon_r}\} \,,$$ (Eq. 10)

where $\mathfrak{I}\{\}$ represents the imaginary part. We calculate the complex dielectric constant of the saturated firn
layer in Eq. 8 and Eq. 10 using the empirically derived models described in Tiuri et al., (1984). We set the
volumetric fraction of meltwater to $m_v$=1%. We set the depth of the water-saturated firn layer in Eq. 7 to
$d$=1 m. These values are consistent with typical lower frequency (e.g., 37 GHz, 13.4 GHz, 19 GHz) passive
(e.g., Mote, et al. 1995; Abdalati and Steffen, 1997; Ashcraft and Long, 2006) and active (e.g., Hicks and
Long, 2011) microwave algorithms used to detect seasonal surface melting over the GrIS. Using the results
of Eq. 7, 8, 9, and 10, we calculate the firn saturation parameter threshold at $\xi_T$=0.1.
The first step in our adapted empirical algorithm is to map the extent of the percolation facies. For
each rSIR grid cell within the ice sheet-masked extent of the GrIS, we smooth the corresponding $T_V^B$ time
series (1 April 2015 - 31 March 2019) using a 14-observation (1 week) moving window. We extract the
minimum vertically-polarized L-band brightness temperature ($T_{V,min}^B$), and the maximum vertically-polarized
L-band brightness temperature ($T_{V,max}^B$). We set the physical temperature of the water-saturated firn layer
to $T$=273.15 K, and the transmission angle to $\theta$=40°. We then calculate the firn saturation parameter ($\xi$)
using Eq. 2. If the calculated firn saturation parameter exceeds the firn saturation parameter threshold, the
rSIR grid cell is converted to a binary parameter to map the total extent of the percolation facies.
We note that smoothing $T_V^B$ time series will mask brief low-intensity seasonal surface melting that
occurs in the high-elevation (> ~2500 m) percolation facies, where seasonal meltwater is rapidly refrozen
within the colder snow and firn layers (e.g., Fig. 4d). Thus, the calculated firn saturated parameter will not
exceed the firn saturation parameter threshold, and these rSIR grid cells will be excluded from the algorithm.
The exclusion of rSIR grid cells in the high-elevation percolation facies is not expected to have a significant
impact on our results as our algorithm targets rSIR grid cells in areas that experience intense seasonal
surface melting. The exclusion of rSIR grid cells will, however, slightly underestimate the mapped
percolation facies extent.
The second step in our adapted empirical algorithm is to map the extent of perennial firn aquifer,
ice slab, and perched firn aquifer areas over the percolation facies. For each rSIR grid cell within the
mapped percolation facies extent, we normalize the corresponding $T_V^B$ time series (1 April 2015 - 31 March
2019) using Eq. 5 ($T_{V,N}^B(t)$). We then extract the initial normalized vertically-polarized L-band brightness
temperature, ($T_{V,N}^B(t_{max})$) and the final normalized vertically-polarized L-band brightness temperature
($T_{V,N}^B(t_{min})$), and partition $T_{V,N}^B(t)$ on the time interval $t \in [t_{max}, t_{min}]$. We smooth $T_{V,N}^B(t \in [t_{max}, t_{min}])$
using a 56-observation (4 week) moving window. The sigmoid fit is then iteratively applied using Eq. 6.



Smoothing reduces the chi-squared error statistic when fitting $T_{V,N}^B\big(t \in [t_{max},\, t_{min}]\big)$ to the sigmoid function.
We fix the initial normalized vertically-polarized L-band brightness temperature at $T_{V,N}^B(t_{max})$=0.99, which
provides a uniform parameter space in which the refreezing rate parameter ($\zeta$) can be analyzed. Variability
in $T_{V,N}^B(t_{max})$ is controlled by the volumetric fraction of meltwater within the upper snow and firn layers of
the percolation facies, and is accounted for in the firn saturation parameter ($\xi$), which is analyzed separately.
$T_{V,N}^B\big(t \in [t_{max},\, t_{min}]\big)$ iteratively fit to the sigmoid function converge quickly (i.e., algorithm iterations I $\in$ [5,
15]), and observations are a good fit (i.e., chi squared error statistic is $\chi2 \in [0, 0.1]$).

Using the SMAP-derived $T_{V,N}^B(t_{max})$ and $T_{V,N}^B(t_{min})$, rather than the MODIS-derived initial

normalized vertically-polarized L-band brightness temperature at the surface freeze-up date ($T_{V,N}^B(t_{sfu})$),
and final normalized vertically-polarized L-band brightness temperature at the melt onset date ($T_{V,N}^B(t_{mo})$)
that were used in the empirical algorithm described in Miller et al., 2020 (e.g., Fig. 4), has several
advantages. They key advantage of this approach is that maps can be generated using $T^B$ imagery
collected from a single satellite, which simplifies the adapted empirical algorithm. Another advantage is that
unlike $T^B$ collected at shorter-wavelength thermal infrared  frequencies (e.g., MODIS), $T^B$ collected at
longer wavelength microwave frequencies (e.g., SMAP) is not sensitive to clouds, which eliminates
observational gaps and cloud contamination, and provides more accurate time series partitioning and more
robust curve fitting. The mapped extent of Greenland's perennial firn aquifers generated by our adapted
empirical algorithm and by our empirical algorithm (Miller et al., 2020) are consistent (see Section 3).

We calibrate our adapted empirical algorithm using the AR- and MCoRDS-derived perennial firn

aquifer (2010-2017), ice slab (2010-2014), and perched firn aquifer (2010-2017) detections projected
separately on three NH EASE-Grids 2.0. For each rSIR grid cell with at least one detection, we extract the
corresponding maximum vertically-polarized L-band brightness temperature ($T_{V,max}^B$), the minimum
vertically-polarized L-band brightness temperature ($T_{V,min}^B$), the firn saturation parameter ($\xi$), and the
refreezing rate parameter ($\zeta$), and for each of the extracted SMAP-derived calibration parameters we
calculate the standard deviation ($\sigma$). Similar to Miller et al., 2020, thresholds of ±$2\sigma$ are set for each of the
extracted SMAP-derived calibration parameters in an attempt to eliminate peripheral rSIR grid cells near
the ice sheet edge and near the upper and lower boundaries of each sub-facie, where L-band emissions
can be influenced by morphological features, such as crevasses, superimposed and glacial ice, and
spatially integrated with emissions from rock, land, the ocean, and adjacent percolation facies and wet snow
facies areas. The SMAP-derived calibration parameter threshold intervals extracted from $T_V^B$ time series
that we use to map perennial firn aquifer, ice slab, and perched firn aquifer areas are given in Table 3. We
apply the calibration to each rSIR grid cell within the percolation facies extent. If the extracted SMAP-
derived calibration parameters are within the threshold intervals, the rSIR grid cell is converted to a binary
parameter to map the total extent of each of these sub-facies.





**Table 3.** *SMAP-derived calibration parameter threshold intervals (1 April 2015 - 31 March 2019) used for mapping perennial firn aquifer, ice slab, and perched firn aquifer areas.*

|  | $\xi$ | $T_{V,max}^{B}$ (K) | $T_{V,min}^{B}$ (K) | $\zeta$ |
|---|---|---|---|---|
| **Perennial Firn Aquifers** | $0.2 - 4$ | $200 - 275$ | $180 - 250$ | -0.04 - -0.02 |
| **Ice Slabs** | $0.1 - 2$ | $170 - 260$ | $130 - 240$ | -0.03 - -0.06 |
| **Perched Firn Aquifers** | $0.2 - 1.2$ | $200 - 260$ | $180 - 240$ | -0.03 - -0.04 |

Iteratively applying the sigmoid fit to $T_{V,N}^{B}\left(t \in [t_{max}, t_{min}]\right)$ over perched firn aquifer areas is a source of uncertainty in our adapted empirical algorithm. While the continuous logistic model is reasonable for the majority of exponentially decreasing temporal L-band signatures over the percolation facies, it is not optimal for exponentially decreasing temporal L-band signatures that transition to linearly decreasing on time scales of ~years following the surface freeze-up date. Especially multi-year linearly decreasing temporal L-band signatures over areas where perching occurs following intense seasonal surface melting and shallow water-saturated firn layers persist throughout the following freezing season as well as throughout weaker seasonal surface melting the following melting season (e.g., between ~September 2016 and May 2018 in Fig. 4b). Although $T_{V,N}^{B}\left(t \in [t_{max}, t_{min}]\right)$ over perched firn aquifer areas iteratively fit to the sigmoid function converge quickly (i.e., algorithm iterations $I \in [8, 15]$), and observations appear to be a good fit (i.e., chi squared error statistic is $\chi2 \in [0.06, 0.1]$), simulated sigmoidal curves often asymptotically approach $T_{V,min}^{B}$ too quickly, which underestimates the refreezing rate parameter ($\zeta$) to values outside the SMAP-derived calibration parameter threshold intervals. Perched firn aquifer areas may alternatively be mapped as ice slab areas or percolation facies areas, which will underestimate or overestimate the mapped extent of each of these sub-facies.

Miller et al., 2020 cited significant uncertainty in the SMAP-derived perennial firn aquifer extent as a result of the lack of a distinct temporal L-band signature delineating the boundary between perennial firn aquifer areas and adjacent percolation facies areas. In this study, similar uncertainty exists in the SMAP-derived perennial firn aquifer, ice slab, and perched firn aquifer extents. This uncertainty could, at least in part, be a result of the rSIR algorithm. An rSIR grid cell corresponds to the weighted average of $T_V^B$ over SMAP's antenna footprint (Long et al., 2020). The weighting is the grid cell's spatial response function (SRF), which is ~18 km (i.e. the effective resolution) in diameter. The SRF is centered on the rSIR grid cell. Since the effective resolution (i.e., size of the 3 dB contour of the SRF) is less than the rSIR grid cell spacing, rSIR grid cell SRF's overlap and the grid cells $T_V^B$ values are not statistically independent. This uncertainty, however, could also have a geophysical basis, as it is unlikely that the boundaries between sub-facies (perennial firn aquifers, ice slabs, and perched firn aquifers) as well as between facies (percolation facies, dry snow facies, wet snow facies) are distinct. The thickness of the water-saturated firn layer or ice slab may thin and taper-off at the periphery, and sub-facies and facies may become spatially scattered and





merge together. Over SMAP's ~18 km footprint, spatially integrated L-band emissions may also result in a
smooth transition between temporal L-band signatures.
The limited extent (AR, 15 m x 20 m; MCoRDS, 14 m x 40 m) of the airborne ice-penetrating radar
detections as compared to the rSIR grid cell extent (3.125 km x 3.125 km) and the effective resolution (~18
km) of the SMAP enhanced-resolution $T_V^B$ imagery is also cited in Miller et al., 2020 as a source of
uncertainty in the empirical algorithm. In this study, similar uncertainty exists in our adapted empirical
algorithm. The total rSIR grid cell extent with radargram coverage is less than 2%, which means that ~98%
of the total rSIR grid cell extent with radargram coverage, from which the SMAP-derived calibration
parameter threshold intervals are extracted, is unknown. Calculating the total rSIR grid cell extent where
detections are absent along OIB flight lines and statistically integrating this calculation into the multi-year
calibration technique may help reduce the uncertainty, particularly the significant uncertainty in the
interannual variability, which we have yet to resolve. A sensitivity analysis suggests that even small changes
in any of the SMAP-derived calibration parameter threshold intervals (i.e., several K for $T_{V,min}^B$, and $T_{V,max}^B$,
several tenths of a percentage point for $\xi$, and several hundredths of a percentage point for $\zeta$) can result in
variability in the mapped extents of hundreds of square kilometers, and boundary transitions between
perennial firn aquifer, ice slab, and perched firn aquifer areas. Thus, the mapped extent of each of these
sub-facies of the broader percolation facies should simply be considered an initial result demonstrating the
potential of our adapted empirical algorithm for future work.

## 3.    Results and Discussion

The SMAP-derived maximum vertically-polarized L-band brightness temperature values generated by our
adapted empirical algorithm range from between $T_{V,max}^B$=150 K and 275 K, and the minimum vertically-
polarized L-band brightness temperature values range from between $T_{V,min}^B$=130 K and 250. These values
are consistent with the range of $T_{V,max}^B$ and $T_{V,min}^B$ values given in the temporal L-band signature analysis
(Section 2.4.2; Table 2). Firn saturation parameter values range from between $\xi$=0.1 and 4.0. Refreezing
rate parameter values range from between $\zeta$=-0.09 and -0.01. The lower bound ($\zeta$=-0.09) of the refreezing
rate parameter observed over the percolation facies is significantly higher than the predicted lower bound
($\zeta$=-1) in our example set of simulated sigmoidal curves (black line, Fig. 6).
The SMAP-derived perennial firn aquifer (blue shading), ice slab (cyan shading), perched firn
aquifer (green shading), and percolation facies (purple shading) extents (2015-2019) generated by our
adapted empirical algorithm are shown in Figs. 7a-9a, and are summarized in Table 4. The percolation
facies extent (~5.8 x $10^5$ km$^2$) generated by our adapted empirical algorithm is mapped at elevations
between ~500 m.a.s.l. and 3500 m.a.s.l., and extends over ~32 % of the GrIS extent (~1.8 x $10^6$ km$^2$). The
perennial firn aquifer extent (64,000 km$^2$) is mapped at elevations between ~600 m.a.s.l and 2600 m.a.s.l.,
and extends over ~ 11% of the percolation facies extent and ~4% of the GrIS extent. High $T_{V,max}^B$, $T_{V,min}^B$,

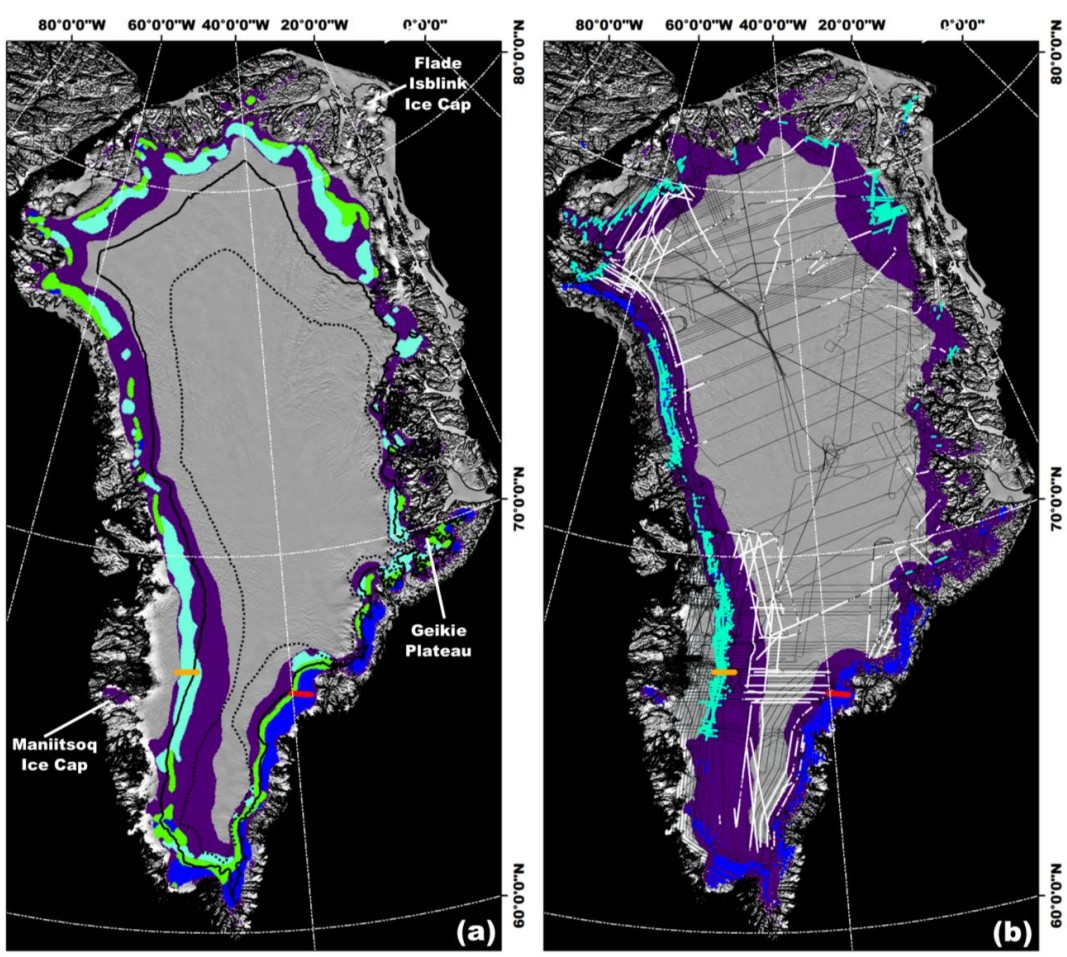

**Figure 7**

*(a) The SMAP-derived perennial firn aquifer (blue shading), ice slab (cyan shading), perched firn aquifer*
*(green shading), and percolation facies (purple shading) extents (2015-2019) generated by the adapted*
*empirical algorithm overlaid on the 2015 MODIS Mosaic of Greenland image map (Haran et al., 2018). The*
*black line is the 2000 m.a.s.l. contour, and the black dotted line is the 2500 m.a.s.l. contour (Howat et al.,*
*2014). (b) The SMAP-derived extents are overlaid with AR- and MCoRDS-derived 2010-2017 perennial firn*
*aquifer (blue shading; Miège et al., 2016), 2010-2014 ice slab (cyan shading; McFerrin et al., 2019), and*
*2012 spatially coherent melt layer (white shading; Culberg et al., 2021) detections along OIB flight lines*
*(black lines). Overlapping perennial firn aquifer and ice slab detections are interpreted as perched firn*
*aquifer areas. The red line is AR radargram profile along perennial firn aquifer transect A-B (Fig. 3a). The*
*orange line is AR radargram profile along ice slab transect C-D (Fig. 3b).*



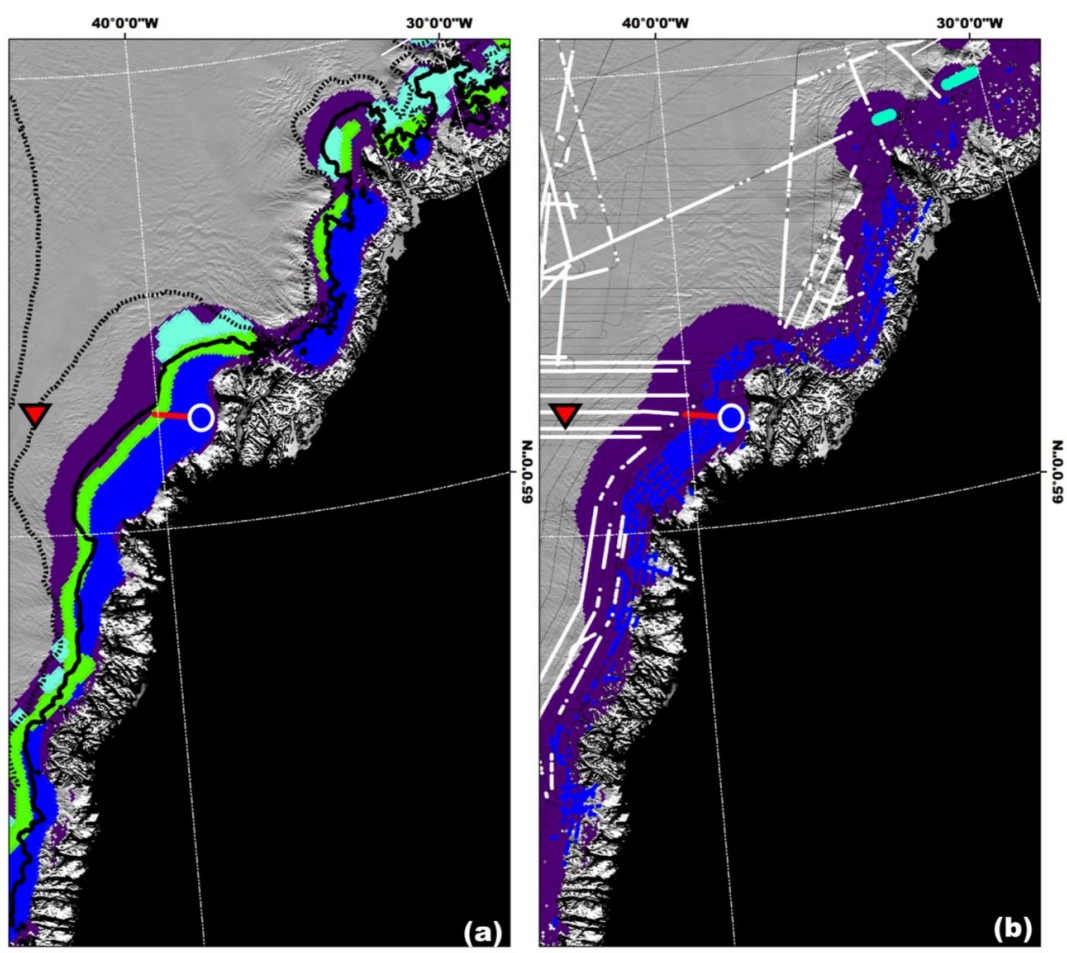

**Figure 8**

*(a) The SMAP-derived perennial firn aquifer (blue shading), ice slab (cyan shading), perched firn aquifer*
*(green shading), and percolation facies (purple shading) extents (2015-2019) generated by the adapted*
*empirical algorithm over south eastern Greenland (Fig. 1c; zoom area in red box) overlaid on the 2015*
*MODIS Mosaic of Greenland image map (Haran et al., 2018). The solid black line is the 2000 m.a.s.l.*
*contour, and the black dotted line is the 2500 m.a.s.l. contour (Howat et al., 2014). (b) The SMAP-derived*
*percolation facies extent is overlaid with AR- and MCoRDS-derived 2010-2017 perennial firn aquifer (blue*
*shading; Miège et al., 2016), 2010-2014 ice slab (cyan shading; McFerrin et al., 2019), and 2012 spatially*
*coherent melt layer (white shading; Culberg et al., 2021) detections along OIB flight lines (black lines).*
*Overlapping perennial firn aquifer and ice slab detections are interpreted as perched firn aquifer areas. The*
*red line is AR radargram profile along perennial firn aquifer transect A-B (Figs. 1; 3a). The blue circle is a*
*perennial firn aquifer area (Figs. 3a; 4a). The red triangle is a high-elevation (~2500 m.a.s.l.) percolation*
*facies area (Figs. 4d).*

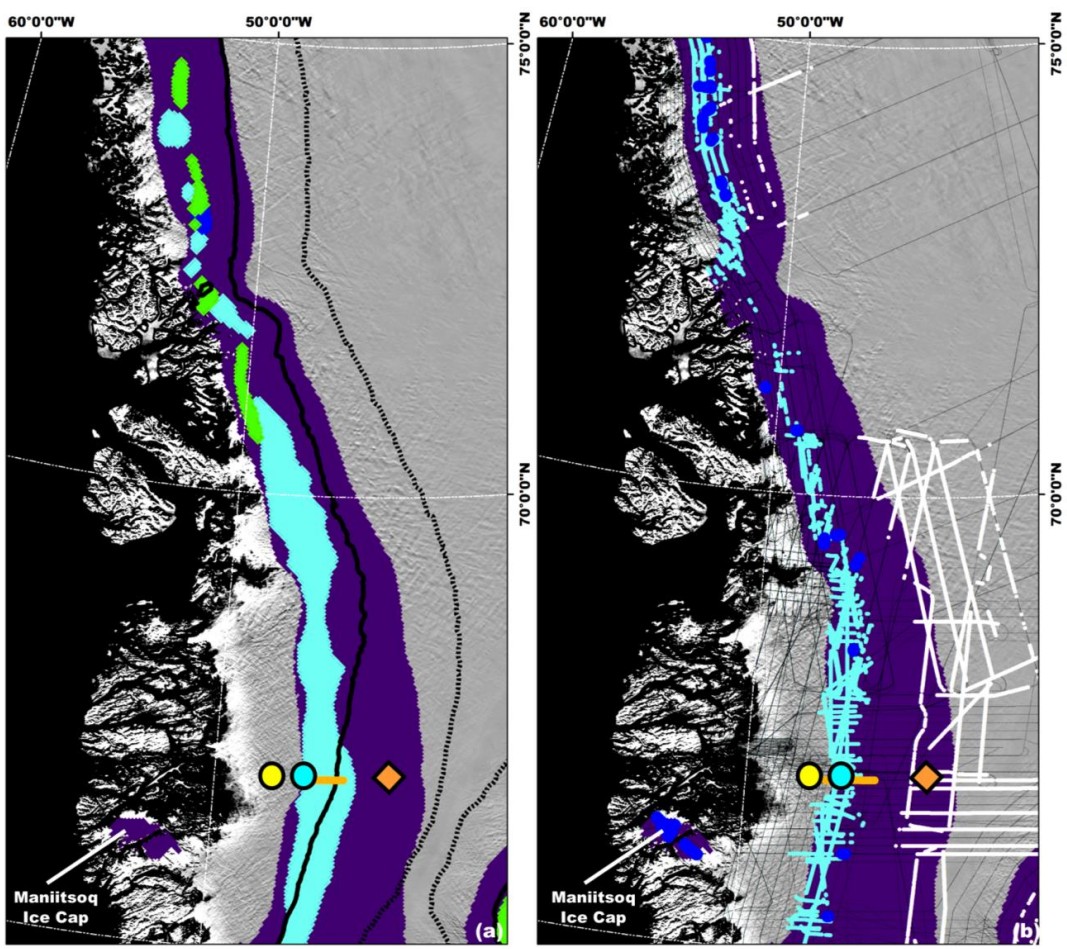

### Figure 9

*(a) The SMAP-derived perennial firn aquifer (blue shading), ice slab (cyan shading), perched firn aquifer*
*(green shading), and percolation facies (purple shading) extents (2015-2019) generated by the adapted*
*empirical algorithm over south western Greenland (Fig. 1c; zoom area in red box) overlaid on the 2015*
*MODIS Mosaic of Greenland image map (Haran et al., 2018). The solid black line is the 2000 m.a.s.l.*
*contour, and the black dotted line is the 2500 m.a.s.l. contour (Howat et al., 2014). (b) The SMAP-derived*
*percolation facies extent is overlaid with AR- and MCoRDS-derived 2010-2017 perennial firn aquifer (blue*
*shading; Miège et al., 2016), 2010-2014 ice slab (cyan shading; McFerrin et al., 2019), and 2012 spatially*
*coherent melt layer (white shading; Culberg et al., 2021) detections along OIB flight lines (black lines).*
*Overlapping perennial firn aquifer and ice slab detections are interpreted as perched firn aquifer areas. The*
*orange line is AR radargram profile along ice slab transect C-D ( Figs 1; 3b). The cyan circle is a perched*
*firn aquifer area (Figs. 3b; 4b). The orange diamond is a percolation facies area (Fig. 4c). The yellow circle*
*is a superimposed ice area (Fig. 4e).*





$\xi$, and $\zeta$ values within the perennial firn aquifer extent indicates the presence of thicker water-saturated
firn layers with larger volumetric fractions of meltwater that are radiometrically warm during both the melting
and freezing seasons and have extended refreezing rates. The ice slab extent (76,000 km$^2$) is mapped at
elevations between ~800 m.a.s.l and 2700 m.a.s.l., and extends over ~13 % of the percolation facies extent
and ~4 % of the GrIS extent. As compared to perennial firn aquifer areas, decreased $T^B_{V,max}$, $T^B_{V,min}$, $\xi$ and
$\zeta$ values indicates the presence of thinner water-saturated firn layers with lower volumetric fractions of
meltwater that are radiometrically colder and have slightly more rapid refreezing rates. The SMAP-derived
perched firn aquifer calibration parameter intervals are within the perennial firn aquifer and the ice slab
calibration parameter intervals (see Fig. 6, Table 3). The extents of these three sub-facies within the broader
percolation facies are overlapping, such that perched firn aquifers typically represent the upper boundary
of perennial firn aquifer areas that subsequently transition to percolation facies as well as the lower
boundary of ice slab areas that subsequently transition to wet snow facies. However, in several areas in
south and southeastern Greenland, the full progression of facies and sub-facies (dry snow facies -
percolation facies - ice slab - perched firn aquifer - perennial firn aquifer – ablation facies) occur (e.g., Fig.
7a). The perched firn aquifer extent (37,000 km$^2$) is mapped at elevations between ~600 m.a.s.l and 2700
m.a.s.l., and extends over ~30% of perennial firn aquifer extent, ~24% of ice slab extent, ~2% of the
percolation facies extent, and less than ~ 1% of the GrIS extent. Combined together, the total extent
(~140,000 km$^2$) is the equivalent of ~24% of the percolation facies extent and 10% of the GrIS extent. This
increases previous AR- and MCoRDS-derived elevation estimates upslope ~600 m.a.s.l. in both perennial
firn aquifer areas (Miège et al., 2016) and ice slab areas (McFerrin et al., 2019). The highest perennial firn
aquifer, ice slab, and perched firn aquifer elevations (>2500 m.a.s.l.) are mapped in southern Greenland
and on the Geikie plateau, in central eastern Greenland.

**Table 4.** *The SMAP-derived perennial firn aquifer, ice slab, and perched firn aquifer extents (2015-2019)*
*over the percolation facies and the GrIS, and the elevation range at which they are mapped.*

|  | Percolation Facies Extent (%) | Ice Sheet Extent (%) | Elevation Range (m.a.s.l.) |
|---|---|---|---|
| **Perennial Firn Aquifers** | 11 | 4 | 600 – 2600 |
| **Ice Slabs** | 13 | 4 | 800 – 2700 |
| **Perched Firn Aquifers** | 5 | <1 | 600 – 2700 |


Figs. 7b-9b shows perennial firn aquifers (blue shading), ice slabs (cyan shading), and spatially
coherent melt layers (white shading) detected by airborne ice-penetrating radar surveys (2010-2017)
overlaid on the SMAP-derived percolation facies extent (2015-2019). Perched firn aquifer areas are inferred
where perennial firn aquifer and ice slab detections overlap. The SMAP-derived perennial firn aquifer extent
mapped in southern, and south and central eastern Greenland is consistent with the AR- and MCoRDS-



geophysical derived perennial firn aquifer detections (2010-2017), except in north western Greenland
where perched firn aquifers are alternatively mapped. The SMAP-derived ice slab extent mapped in
western, central and north eastern, and northern Greenland is generally consistent with the spatial patterns
of the AR-derived ice slab detections (2010-2014), however, is significantly expanded upslope in each of
these areas. We note that the AR-derived ice slab detections are limited in space and time, particularly in
northern Greenland, with a time interval as large as nine years between the airborne ice penetrating radar
surveys and the SMAP enhanced-resolution $T_V^B$ imagery used in the adapted empirical algorithm (i.e., 2010
to 2019). In western and northern Greenland, the 2015 melting season was especially intense (Tedesco et
al., 2016). And, in northern Greenland, the ablation facies have recently increased in extent (2010-2019;
Noël et al., 2019), and supraglacial lakes have recently advanced inland (2014-2019; Turton et al., 2021),
indicating a likley geophysical basis for the observed upslope expansion. In central and north eastern, and
northern Greenland, perched firn aquifers are often alternatively mapped. Additional smaller ice slab areas
are mapped in south and south eastern (Figs. 9a; 9b) Greenland. The scattered SMAP-derived perched
firn aquifer extent mapped in north western and central eastern Greenland is fairly consistent with the
sparse AR- and MCoRDS-derived perched firn aquifer detections (2010-2017), however, in central western
Greenland (Figs. 9a; 9b) ice slab areas are alternatively mapped. Expansive additional perched firn aquifer
areas are mapped in southern, and south and central eastern Greenland. These areas are often coincident
with spatially coherent melt layer detections, particularly in south eastern Greenland (Figs. 8a, 8b). Neither
perennial firn aquifer, ice slab, nor perched firn aquifer areas are mapped on the Maniitsoq and Flade Isblink
Ice Caps. Over these two small ice caps, L-band emissions spatially integrated with emissions from rock,
land, the ocean, and adjacent percolation facies and wet snow facies areas result in SMAP-derived
calibration parameter values outside the defined intervals for each of these sub-facies.
We infer that the SMAP-derived perched firn aquifer extent represents L-band emissions from: (1)
spatially expansive, relatively shallow water-saturated firn layers with lower volumetric fractions of
meltwater as compared to perennial firn aquifer areas. These shallow water-saturated firn layers transiently
form on top of buried ice slabs, spatially coherent melt layers, or other semi-impermeable layers that have
previously formed within the upper snow and firn layers of the percolation facies, as shown in Figs. 7-8. Or,
(2) spatially scattered deeper water-saturated firn layers with larger volumetric fractions of meltwater (i.e.,
perennial firn aquifers) that are spatially integrated with L-band emissions from adjacent ice slabs,
percolation facies, and/or wet snow facies areas. These areas are observed as shallow water-saturated firn
layers with lower volumetric fractions of meltwater over SMAP's ~18 km footprint (i.e., the effective
resolution). Or, (3) a combination of these englacial firn hydrological features, which is a likely scenario
over many perched firn aquifers areas. This is particularly likely in north western Greenland, where airborne
ice penetrating radar surveys consistently detect perennial firn aquifers; however, the SMAP-derived extent
indicates perched firn aquifer areas (Fig. 7).





1013   Shallow buried supraglacial lakes have recently been identified within the percolation facies of
1014 western, northern, and north and central eastern Greenland using airborne ice penetrating radar surveys
1015 (Koenig et al., 2015) and satellite synthetic aperture radar imagery (Miles et al., 2017; Schröder et al., 2020;
1016 Dunmire et al., 2021). These buried supraglacial lakes are within the SMAP-derived perennial firn aquifer,
1017 ice slab, and perched firn aquifer extents, however, they are not expected to significantly influence L-band
1018 emissions in these areas for two reasons: (1) as compared to SMAP's ~18 km footprint, the mean extent
1019 of buried supraglacial lakes is limited (less than ~1 km$^2$), and they are sparsely distributed in perennial firn
1020 aquifer, ice slab, and perched firn aquifer areas (Dunmire et al., 2021). (2) Supraglacial lakes form during
1021 the melting season as a result of meltwater storage in topographic depressions at the ice sheet surface
1022 (Echelmeyer et al. 1991). Similar to subglacial lakes (Jezek et al., 2015) and perennial firn aquifers (Miller
1023 et al., 2020), supraglacial lakes represent radiometrically cold subsurface meltwater reservoirs. Upwelling
1024 L-band emissions from deeper firn layers, glacial ice, and the underlying bedrock are effectively blocked by
1025 high reflectivity and attenuation at the impermeable layer-lake bottom interface. This results in a low
1026 observed $T_V^B$ at the upper surface of meltwater stored within supraglacial lakes. During the freezing season,
1027 the upper surface of meltwater stored within supraglacial lakes refreezes and forms a partial or solid-ice
1028 cap that is sometimes buried by snow accumulation (Koenig et al., 2015). Airborne ice penetrating radar
1029 surveys in April and May between 2009 and 2012 suggest the mean depth to the upper surface of meltwater
1030 stored within buried supraglacial lakes is ~2 m (Koenig et al., 2015). As previously noted, over perennial
1031 firn aquifer, ice slab, and perched firn aquifer areas, L-band emissions from the radiometrically warm upper
1032 snow and firn layers decrease on variable time scales during the freezing season as embedded ice
1033 structures slowly refreeze at increased depths below the ice sheet surface and induce strong volume
1034 scattering (Rignot et al., 1993; Rignot 1995). $T_V^B$ can decrease by as much as ~50 K during the freezing
1035 season (e.g., Fig. 4a), representing the descent of the upper surface of stored meltwater by ~tens of meters
1036 (Miège et al., 2016). However, over buried supraglacial lakes, L-band emissions from the refreezing partial
1037 or solid-ice cap, which is smooth relative to the L-band wavelength (~21 cm), induce surface scattering. As
1038 a result, $T_V^B$ decreases over buried supraglacial lakes are negligible. Thus, over SMAP's ~18 km footprint,
1039 water-saturated firn layers dominate L-band emissions over the percolation facies of the GrIS.
1040   The SMAP-derived perennial firn aquifer extent (~64,000 km$^2$) generated by our adapted empirical
1041 algorithm and the multi-year (2010-2017) calibration technique is consistent with the extent (~66,000 km$^2$)
1042 generated by the previously developed empirical algorithm and the single-coincident year (2016) calibration
1043 technique described in Miller et al., 2020. The SMAP-derived perennial firn aquifer extent is generally
1044 consistent with previous C-band (5.3 GHz) satellite radar scatterometer-derived perennial firn aquifer
1045 extents mapped using the Advanced SCATterometer (ASCAT) on the European Organization for the
1046 Exploitation of Meteorological Satellites (EUMETSAT) Meteorological Operational A (MetOp-A) satellite
1047 (2009-2016, ~52 000-153 000 km$^2$; Miller, 2019), and the Active Microwave Instrument in radar
1048 scatterometer mode (ESCAT) on ESA's European Remote Sensing (ERS) satellite series (1992-2001,
1049 ~37 000-64 000 km$^2$; Miller, 2019) as well as the C-band (5.4 GHz) synthetic aperture radar-derived extent



1050 mapped using ESA's Sentinel-1 satellite (2014-2019, ~54 000 km$^2$; Brangers et al., 2020). The exception

1051 is the ASCAT-derived perennial firn aquifer extent (2012-2013, ~153,000 km$^2$; Miller et al., 2019) mapped

1052 following the anomalous 2012 melting season (Nghiem et al., 2012) in which significant changes in the

1053 dielectric and geophysical properties that influence radar backscatter and the temporal C-band signatures

1054 occurred. The unreasonably expansive (i.e., more than twice the mean) mapped extent is a result of

1055 ASCAT'S shallow (~several meters) C-band penetration depth (Jezek et al., 1994), and the simple

1056 threshold-based algorithm that was not calibrated for an extreme melting season that included saturation

1057 of the upper snow and firn layers of the dry snow facies and percolation facies with relatively large

1058 volumetric fractions of meltwater (Miller et al., 2019). Water-saturated firn layers had extended refreezing

1059 rates, however, seasonal meltwater was not stored at depth. Spatially coherent melt layers were

1060 alternatively formed in many of the mapped areas (Culberg et al., 2021). The SMAP-derived ice slab extent

1061 (~76,000 km$^2$) is also consistent with previous AR-derived ice slab extents (2010-2014, ~64,800 km$^2$-69,400

1062 km$^2$; McFerrin et al., 2019).

1063  Although we simply consider our mapped extents a high-probability area for preferential formation,

1064 the maps generated by our adapted empirical algorithm and the multi-year (2010-2017) calibration

1065 technique for individual years suggest interannual variability in perennial firn aquifer, ice slab, and perched

1066 firn aquifer extents, which is summarized in Table 5. Our results demonstrate reasonable sensitivity to

1067 variability in the dielectric and geophysical properties that influence the radiometric temperature and

1068 temporal L-band signatures, even during the extreme 2015 melting season (Tedesco et al., 2016).


1070 **Table 5** *The SMAP-derived perennial firn aquifer, ice slab, and perched firn aquifer extents (2015-2019).*

| | Perennial Firn Aquifer Extent (km$^2$) | Ice Slab Extent (km$^2$) | Perched Firn Aquifer Extent (km$^2$) |
|---|---|---|---|
| **2015-2019** | 66,000 | 76,000 | 37,000 |
| **2015-2016** | 63,000 | 23,000 | 17,000 |
| **2016-2017** | 69,000 | 48,000 | 38,000 |
| **2017-2018** | 73,000 | 27,000 | 20,000 |
| **2018-2019** | 70,000 | 38,000 | 26,000 |










## 5 Summary and Future Work

L-band satellite microwave sensors – including NASA's L-band SMAP mission - represent a relatively new Earth-observation tool that has exceptional capabilities for cryospheric applications. Especially, mapping englacial and subglacial hydrological features at depths of ~tens to hundreds of meters beneath the surface of Earth's polar ice sheets. In this study, for the first time, we have exploited this capability and demonstrated the novel use of the L-band microwave radiometer on NASA's SMAP satellite for mapping perennial firn aquifers, ice slabs, and perched firn aquifers together as a continuous system over the percolation facies of the GrIS. We have also demonstrated that SMAP enhanced-resolution L-band $T_V^B$ imagery can effectively resolve percolation facies features that are not effectively resolved in conventionally processed SMAP L-band $T_V^B$ imagery (e.g., Fig. 1). We have adapted our previously developed empirical algorithm (Miller et al., 2020) by expanding our analysis of spatiotemporal differences in SMAP enhanced-resolution $T_V^B$ imagery and temporal L-band signatures over the GrIS. We have used this analysis to derive a firn saturation parameter from a simple two-layer L-band geophysical-brightness temperature model. And, we have used the firn saturation parameter to map the extent of the percolation facies. We have found that by correlating maximum and minimum $T_V^B$ values, the firn saturation parameter, and the refreezing rate parameter with perennial firn aquifer, ice slab, and perched firn aquifer detections identified via NASA's OIB campaigns that we can calibrate our previously developed empirical algorithm (Miller et al., 2020) to map plausible extents.

We note that significant uncertainty exists in the mapped extents as a result of (1) correlating the SMAP-derived parameters with airborne ice-penetrating radar detections that are not coincident in time, (2) the lack of a distinct temporal L-band signature delineating the boundary between each of the mapped sub-facies within the broader percolation facies, and (3) the much more limited extent of the airborne ice-penetrating radar detections as compared to the rSIR grid cell extent, as well as the effective resolution of the SMAP enhanced-resolution $T_V^B$ imagery. Additional uncertainty exists in the perched firn aquifer extent as a result of fitting L-band signatures to the continuous logistic model, which is not optimal for these specific sub-facies.

Miller et al., (2020) normalized SMAP enhanced-resolution $T_V^B$ time series and converted the exponential rate of $T_V^B$ decrease over perennial firn aquifer areas to a binary parameter to map extent. In this study, we have converted the SMAP-derived parameters to binary parameters to map the extent of perennial firn aquifer, ice slab, and perched firn aquifer areas. Moreover, we have included additional analysis of the spatiotemporal differences in maximum and minimum $T_V^B$ values, the firn saturation parameter, and the refreezing rate parameter. We have shown that spatiotemporal differences in the SMAP-derived parameters are consistent with our assumption of spatiotemporal differences in the englacial hydrology and thermal characteristics of firn layers at depth. Particularly, our assumption that latent heat release influences temporal L-band signatures within the percolation facies of the GrIS. This includes continuous latent heat release via the slow refreezing of the deeper firn layers in perennial and perched firn aquifer areas that are saturated with large volumetric fractions of meltwater. And, latent heat release that





occurs throughout the percolation facies via more rapid refreezing of seasonal meltwater by the descending
winter cold wave, and the subsequent formation of embedded ice structures, including ice slabs and
spatially coherent melt layers, within the upper snow and firn layers.

Future work will focus on simulating maximum and minimum $T_V^B$, the firn saturation parameter, and

the refreezing rate parameter as well as temporal L-band signatures observed over perennial firn aquifer,
ice slab, and perched firn aquifer areas within the percolation facies of the GrIS for a wide range of
geophysical properties. Significant interannual variability in the dielectric and geophysical properties that
seasonally influence the radiometric temperature and temporal L-band signatures can occur, particularly
following extreme melting seasons, such that it is critical that these properties are understood and
considered in any given year. To better interannual variability as well as other geophysical properties, we
will interpret our results together with climatological parameters, such as snow accumulation, liquid water
content, temperature, and surface mass balance, and over the GrIS simulated using the Regional
Atmospheric Climate Model (RACMO2.3p2; Noël et al., 2018). Additionally, we will simulate the distinct
temporal L-band signatures observed over spatially coherent melt layers in the upper snow and firn layers
of the dry snow facies and percolation facies of the GrIS recently identified via MCoRDS flown by NASA's
OIB campaigns (Culberg et al., 2021) following the anomalous 2012 melting season (Nghiem et al., 2012)
and as well as explore the potential for mapping the extent of these near-surface englacial hydrological
features using satellite L-band microwave radiometry. Nghiem et al., (2003) previously demonstrated
mapping spatially coherent melt layers that were formed following the anomalous 2002 melting season
(Steffen et al., 2004) using similar signatures observed in Ku-band radar backscatter time series collected
by the SeaWinds radar scatterometer that was flown on NASA's QuikSCAT satellite (Tsai et al., 2000).
Combining multi-layer depth-integrated L-band geophysical-brightness temperature models (e.g., Jezek et
al., 2015) that include embedded ice structure parametrizations (e.g., Jezek et al., 2018) with models of
depth-dependent geophysical parameters can lead to an improved understanding of the extremely complex
and very poorly described physics controlling L-band emissions over the percolation facies of the GrIS. For
L-band emissions over perennial firn aquifer, ice slab, perched firn aquifer, and spatially coherent melt layer
areas, the key geophysical parameters include atmospheric temperature forcing, physical temperature
versus depth, latent heat, snow accumulation, the volumetric fraction and depth of meltwater, and the
volumetric fraction and geometric configuration of embedded ice structures. The development of more
sophisticated empirical algorithms that incorporate multi-layer depth-integrated L-band geophysical-
brightness temperature models that are constrained by in situ measurements can help reduce the
significant uncertainty in the current mapped extents, and provide more accurate boundaries delineating
each of these sub-facies within the broader percolation facies that can be used to quantify variability in
extent. As Greenland's climate continues to warm, and seasonal surface melting increases in extent,
intensity, and duration, quantifying the possible rapid expansion of each of these sub-facies using satellite
L-band microwave radiometry has significant implications for understanding ice sheet-wide variability in



englacial firn hydrology resulting in meltwater-induced hydrofracturing and accelerated ice flow as well as high-elevation run-off that can impact the mass balance and stability of the GrIS.

The results presented in this study demonstrate the outstanding potential of L-band satellite microwave sensors for mapping englacial firn hydrological features within the percolation facies of the GrIS that can be extended to forthcoming satellite missions, such as the NASA-ISRO SAR mission (NISAR), ESA's Copernicus Imaging Microwave Radiometer (CIMR) mission, ESA's Copernicus Radar Observation System for Europe in L-band (ROSE-L) mission, and candidate missions, such as ESA's Earth Explorer 10 Cryorad mission.

## Data Availability

SMAP enhanced-resolution L-band $T_V^B$ imagery (2015-2019) have been produced as part of the NASA Science Utilization of SMAP project and are available at https://doi.org/10.5067/QZ3WJNOUZLFK (Brodzik et al., 2019). The NASA MEaSUREs Greenland Ice Mapping Project (GIMP) Land Ice and Ocean Classification Mask, Version 1, is available at https://doi.org/10.5067/B8X58MQBFUPA (Howat, 2017), and the Digital Elevation Model, Version 1, is available at https://nsidc.org/data/nsidc-0645/versions/1 (Howat et al., 2015). The coastline data are available from GSHHG – A Global Self-consistent, Hierarchical, High-resolution Geography Database https://doi.org/10.1029/96JB00104 (Wessel and Smith, 1996). Ice surface temperature imagery (2015-2019) have been produced as part of the Multilayer Greenland Ice Surface Temperature, Surface Albedo, and Water Vapor from MODIS V001 data set and are available at https://doi.org/10.5067/7THUWT9NMPDK (Hall and DiGirolamo, 2019). OIB AR- and MCoRDS-derived perennial firn aquifers detections (2010-2017) are available at https://arcticdata.io/catalog/view/doi:10.18739/A2985M (Miège et al., 2016). OIB AR-derived ice slab detections (2010-2014) are available at https://doi.org/10.6084/m9.figshare.8309777 (McFerrin et al., 2019). OIB AR-derived spatially coherent melt layer detections (2017) are available at (https://doi.org/10.18739/A2736M33W) (Culberg et al., 2021). OIB AR L1B Geolocated Radar Echo Strength Profiles, Version 2, are available at, https://doi.org/10.5067/0ZY1XYHNIQNY (Paden et al., 2018). NASA MEaSUREs MODIS Mosaic of Greenland (MOG) 2015 Image Map, Version 2, is available at https://nsidc.org/data/NSIDC-0547/versions/2 (Haran et al., 2018). SMAP-derived perennial firn aquifer, ice slab, and perched firn aquifer extents are available from JZM upon request.

## Author Contributions

JZM initiated the study, adapted the empirical model, performed the analyses, and wrote the manuscript. RC processed and interpreted the OIB AR radargram profiles. RC and DMS provided the spatially coherent melt layer detections. All authors participated in discussions and reviewed manuscript drafts.





## Competing Interests

The authors declare that they have no conflict of interest.'

## Financial Support

JZM, DGL, and MJB are supported by the NASA SMAP science team (no. 80NSSC20K1806), and by the NASA Cryospheric Science Program (no. 80NSSC18K1055 and no. 80NSSC21K0749) under grants to the University of Colorado and Brigham Young University. RC is supported by a National Defense Science and Engineering Graduate Fellowship. RC and DMS are supported in part by NASA (no. NNX16AJ95G and NSF (no. 1745137). CAS is supported by the NASA Headquarters Cryospheric Science Program. We acknowledge the use of data from CReSIS generated with support from the University of Kansas, NASA Operation IceBridge grant NNX16AH54G, NSF grants ACI-1443054, OPP-1739003, and IIS-1838230, Lilly Endowment Incorporated, and Indiana METACyt Initiative.

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
