# Peer review of "An empirical algorithm to map perennial firn aquifers, ice slabs, and perched firn aquifers within the Greenland Ice Sheet using satellite L-band microwave radiometry"

_The Cryosphere, 2021_

## Author Response (AR1)

**Author Response to Reviewer #1**

*The paper comprehensively describes the method for analysis and the modeling exercise. The authors present the results and fairly point out the sources of uncertainty. I find this paper a valuable contribution to our knowledge of using L-band radiometry for mapping of subsurface features of the GrIS. Nevertheless, I believe the paper could be much more concise. Thus, in addition to addressing the scientific comments and questions, I strongly suggest that authors try to shorten the text which I hope leads to fewer redundancies and better readability.*

**Note: the editor initially asked the authors to respond to the reviewers' comments before revising the paper. So, the authors first described the intended revisions. The authors have included these original comments bolded in black. However, the final changes that were made to the manuscript are in red.**

**The authors thank Reviewer #1 for the positive response and detailed review of a long manuscript. We will significantly shorten the manuscript and will remove the redundancies to improve the readability. We have provided detailed responses to the insightful technical questions.**

**The authors took this comment very seriously, and spent a great deal of time shortening the paper to improve the readability. All of the text, and all of the figure captions were revised. There shouldn't be any redundancies.**

*General Comments and Technical Questions*

*(1) L 509: The closer we are to the Brewster angle, the less sensitive the TBV measurements are to the snowpack dielectric properties. In other words, TBV observations are more sensitive to the subnivean layer properties. With this in mind and that the SMAP observations are at 40°, could you describe the uncertainties introduced to your analysis when considering an ice surface temperature of -1°C and linking them to the TBV measurements?*

**Great question!**

**The authors struggled with this uncertainty while developing the original algorithm (Miller et al., 2020), and then made the decision not to use the MODIS ice surface temperature measurements in the adapted algorithm (this paper), although they are shown in Figure 4. Frequency differencing to look for surface and subsurface meltwater is a good theoretical concept, however, it is not ideal when applied to satellite data. The 'uncertainty gap' between the lack of sensitivity to the dielectric changes at the Brewster angle (i.e. surface melting) in the SMAP TB data and assuming melt when the MODIS ice surface temperature has potentially not actually reached the melting point (i.e., -1˚C) was easily observed when partitioning L-band TB time series. MODIS ice surface temperature measurements are also significantly influenced by the presence of**

clouds. There were often significant surface melting events in the SMAP TB data after surface freeze-up as detected by the MODIS ice surface temperature measurements. The poorly partitioned L-band time series that sometimes included sharp increases were then fit to the sigmoidal curves, which led to uncertainty in the mapping. Changes in the curve fitting values (i.e., rate of TB decrease) could change the mapped boundaries by hundreds of square kilometers.

The authors alternatively used maximum and minimum SMAP TB values to partition L-band TB time series.

**From the original paper (L813-819)**

They key advantage of this approach is that maps can be generated using TB imagery collected from a single satellite, which simplifies the adapted algorithm. Another advantage is that unlike TB collected at shorter-wavelength thermal infrared frequencies (e.g., MODIS), TB collected at longer wavelength microwave frequencies (e.g., SMAP) are not sensitive to clouds, which eliminates observational gaps and cloud contamination, and provides more accurate time series partitioning and more robust curve fitting.

*(2) Can you explain more what do you expect to be the effect of using multi-angle TB measurements on your analysis results?*

Multi-angle L-band TB measurements (e.g., SMOS) introduce a tool to help differentiate how volume and surface scattering components contribute to the observed TB at the surface of the ice sheet, and how this contribution changes with time. For L-band emissions in the percolation facies, the dominant control on volume scattering is embedded ice structures (i.e., ice pipes and lenses), and the dominant control on surface scattering is the volumetric fraction of meltwater within the water-the saturated layer, or possibly the surface roughness of the solid-ice layer (i.e., ice slab). Analysis of multi-angle TB measurements may provide an increased understanding about refreezing processes at depth within the water-saturated firn layer. For example, if a water-saturated firn layer refreezes as an ice slab, the surface scattering component may dominate. However, if a deep perennial firn aquifer exists at depth, volume scattering from the embedded ice pipes and lenses in the overlaying firn may dominate.

*(3) The authors have used enhanced-resolution L-band imagery which includes using multiple satellite overpasses over an area for improving spatial resolution. Given the fact that surface melt could be significantly different between two local overpasses at the similar local times but different days, would you expect to see a difference if you'd repeat your work with the original lower resolution SMAP data?*

Surface melting events over perennial firn aquifer and ice slab areas typically saturate deeper firn layers with meltwater relatively quickly (i.e., days-weeks), and

deeper firn layers typically remain water-saturated throughout the melting season (i.e., months). This results in a superimposed signal – where the rapid daily temperature cycling signal is superimposed over the slowly-varying seasonal signal from the deeper water saturated firn layers (e.g., Figs. 1 & 2). Our algorithm detects the slowly varying seasonal signal from the deeper water saturated firn layers. The maps generated using the enhanced resolution TB data will have a higher effective resolution gridding as compared to the coarse-resolution TB data, which will refine the boundaries of the area mapped. This is the key difference we've observed when applying the algorithm to the different TB data sets.

*(4) In the last section, when talking about future work and potential ideas to follow, it is perhaps good to shed light on the usefulness of other satellites' data or future missions. As an example, ESA's ROSE-L mission could provide us with valuable lower frequency active measurements of the ice sheets.*

**The authors will add a short discussion on the potential advantages of using an active microwave sensor (e.g., ROSE-L or NISAR) or a combined active/passive technique (e.g., SMAP, SMOS, CIMR, Cryorad). The advantage of active microwave sensors is the improvement in spatial resolution. An advantage a combined active-passive microwave technique is the differing sensitivity to physical temperature.**

**The authors removed the reference to forthcoming sensors in the revised manuscript to keep the text tightly focused on SMAP and the current mapping technique.**

***More detailed comments:***

*L 40—44: The sentence is too long. Please break it down to two or three sentences for improved readability. "An empirical algorithm previously developed to map the extent of Greenland's perennial firn aquifers via fitting exponentially decreasing temporal L-band signatures to a set of sigmoidal curves is recalibrated to also map the extent of ice slab and perched firn aquifer areas using airborne ice-penetrating radar surveys collected by NASA's Operation Ice Bridge (OIB) campaigns (2010-2017)." → ""An empirical algorithm was previously developed to map the extent of Greenland's perennial firn aquifers via fitting exponentially decreasing temporal L-band signatures to a set of sigmoidal curves. This algorithm is recalibrated to also map the extent of ice slab and perched firn aquifer areas using airborne ice-penetrating radar surveys collected by NASA's Operation Ice Bridge (OIB) campaigns (2010-2017)."*

**Thank you. Sentence wording will be changed in the text to your suggestion.**

**L36-40 We use an empirical algorithm previously developed to map the extent of Greenland's perennial firn aquifers via fitting exponentially decreasing temporal L-band signatures to a set of sigmoidal curves. This algorithm is recalibrated to**

**also map the extent of ice slab areas using airborne ice-penetrating radar surveys collected by NASA's Operation Ice Bridge (OIB) campaigns (2010-2017***).**

*L 59: "~tens" → "approximately tens"*

**This change will be made.**

**The symbol ~ was removed throughout the revised manuscript.**

*L 110—144: It is just a suggestion. Can the authors include a table that help summarize and explain the formation features and relative TBs of each of the three types of firn structures they discuss in the introduction? For example the rows could be "percolation facies areas", "ice slabs", "perennial firn aquifer areas". The introductory material is written well; nevertheless, it is a bit long for an introduction section in a paper.*

**The authors careful considered this suggestion – as it is a good one. However, a similar table with values derived from the analysis is already included in the Methods section (Table 2). The authors will significantly shorten the introduction.**

**The authors shortened the introduction by several paragraphs.**

*L 175—197: This part can be shortened as it goes into details which best fit in the "Methods" section.*

*L 272-277: The statement given in these lines is basically the same as in lines 175 to 183. Please remove these redundancies.*

L 175-197, and L 272-277 will be combined and significantly shortened in a paragraph at the beginning of the methods section - which will address the previous two comments.

*L 209—214: I see this as a redundancy to the same information provided in the "Introduction" section.*

*L 218: "since the beginning of the satellite era" unnecessary. Can be omitted.*

*L 233—234: "Deep enough to directly detect the upper surface of stored meltwater over the entire depth range mapped by airborne ice-penetrating radar surveys over the GrIS." This sentence is grammatically incorrect. Please revise it.*

*L 236: The beginning of Section 2.2 contains introductory information about SMAP. It is best to include such information in Section 2.1. where you first talk about SMAP and using its passive observations.*

**Section 2.1 will be removed to shorten the manuscript – which addresses the previous four comments.**

**Section 2.1 was removed from the revised manuscript.**

*L 323: "Enhanced-resolution (3.125 km)". This expression potentially created a misunderstanding in reader's mind. The grid is 3.125 km while the actual spatial resolution is at best ~18 m. Please revise the wording to avoid this misunderstanding. The same comment is true for the caption of Fig. 1.*

**The authors agree that it is important to note the difference between the gridding and the effective resolution. The wording in Fig. 1 and Fig. 2 will be changed to:**

**Conventionally processed (25 km gridding, ~30 km effective resolution)**

**Enhanced-resolution (3.125 km gridding, ~18 km effective resolution)**

**The wording in the revised manuscript was changed to:**

**L1133-1134** (a) Gridded (25 km gridding, 30 km effective resolution), and (b) enhanced-resolution (3.125 km gridding, 18 km effective resolution)

*L 326: "(Brodzik et al., 2019) "This reference does not seem to be needed here.*

**Brodzik et al., 2019 is an enhanced resolution data citation for the figures, which is required by the The Cryosphere. For clarity, the reference will be changed in both Fig. 1 and Fig. 2 to:**

**L-band TVB imagery (Brodzik et al., 2019)**

*L 327: There is no panel (c) in Fig.2. It seems that the statement for panel (c) is copied and pasted in the caption for Fig 2.*

*General comment about figures and their captions: The figure captions are too long and they seem to go beyond a general description necessary to read the maps over to detailed discussion of the content.*

**The figure captions will be significantly shortened, and the more descriptive text will be removed.**

**All the figure captions were revised and significantly shortened.**

**See lines L1131-1233 in the revised manuscript.**

*Figure 3: Please explicitly write the unit of the values in the colorbars next to the radargram profiles (panels (a) and (b)).*

**This will be corrected. The colorbar/units were clipped off by mistake.**

**Corrected.**

*L 450: Much of the text after Table 2 can be summarized. Thanks to Table 2, there is not a strong need of writing down the same numbers within the text.*

**Good observation! Text will be revised and shortened.**

**Revised and shortened. See LL 285-363.**

*L 459 – 469: At this point we are far away from the "Introduction" and you include unnecessary background information including about the methods for mapping Greenland's ice facies. Please remove these statements and simply explain the method you have developed for this purpose based on L-band measurements.*

**Section 2.4.1 (and these lines) will be removed to shorten the manuscript.**

**These lines were removed.**

*L 1130—1133: Please keep the Summary and Future Work section free from material which are supposed to be in the "Introduction".*

**These lines will be removed.**

**These lines were also removed.**

**Author Response to Reviewer #2**

*The authors present a very thorough study of mapping perennial firn aquifers and ice slabs using satellite L-band microwave radiometry. The manuscript is rather technical but extremely well thought out and provides robust results and a sophisticated new method that will be of benefit to the community in further entangling the liquid water storage on the Greenland ice sheet. The figures are clear and the placing of the sections obvious.*

*As I am not an expert on the observational techniques and algorithms used, my main comments will be on the structure of the manuscript, and I hope the other reviewers will provide more detailed comments on the methods used. My main comment is that the manuscript is too long and too technical, especially the abstract, introduction and method sections. For the manuscript to become at all readable to a neutral reader with no existing knowledge on the topic, these sections should be significantly cleaned up. For instance, the introduction (and to a smaller extent the abstract) includes lots of technical explanations of the techniques used, while it should just serve as introducing the perennial aquifers, ice slabs and perched aquifers, their importance for cryospheric studies and a brief outline of the manuscript. Several paragraphs could basically be omitted or combined and number of pages considerably cut down. If the authors are able to tighten up the manuscript I recommend publication, solely on the basis that even though my comments above are critical, the overall results and conclusions are clear and convincing.*

**Note: the editor initially asked the authors to respond to the reviewers' comments before revising the paper. So, the authors first described the intended revisions. The authors have included these original comments bolded in black. However, the final changes that were made to the manuscript are in red.**

**The authors want to thank Reviewer #2 for taking the time to review a paper outside of their area of expertise. Especially, given technical nature of the paper. The authors can appreciate Reviewer #2's point of view on the technical content, but since our observational techniques are completely new, we consider the technical content to be an essential part the paper. The technical content provides details of the previously undescribed observational techniques that enable us to derive the algorithm, and support the conclusions drawn. Thus, we do not wish to reduce the critical technical detail. As we move forward with using the derived algorithm, we will focus on the science and not include this level of technical detail in future papers. As advised by Reviewer #2, we will remove repetitions to shorten the text to improve the overall readability of the paper.**

*Below I note a few things in abstract, intro and summary that serve as a basis to how the manuscript could be shortened:*

*P1, l20: No note on what perennial means. One of the characteristics of the perennial firn aquifers is that they last through winter; they are perennial. I do not see this*

*explained anywhere, while this is important for the reader to understand the importance of this phenomenon.*

**Although it is true that they last throughout winter, they are also present in the spring (prior to melting), summer, and autumn (after melting), making them year-round ice sheet features or 'perennial'. The authors will add the term 'year-round' for clarity.**

**L71-73 High snow accumulation in perennial firn aquifer areas thermally insulates water-saturated firn layers from the cold atmosphere allowing seasonal meltwater to be stored in liquid form year-round if the overlying seasonal snow layer is sufficiently thick (Kuipers Munneke et al., 2014).**

*P1, l40-43: Drop the technicalities. Just note: "An recalibrated empirical algorithm is used to map the extent of aquifers".*

**As noted above, given this paper is specifically on the algorithm, and the authors would like to keep the technical details in the abstract. The technical description is also consistent with our previous paper on mapping perennial firn aquifers using SMAP (also in The Cryosphere, https://tc.copernicus.org/articles/14/2809/2020/tc-14-2809-2020.pdf.) The sigmoidal curves are widely used throughout science and engineering; however, this is the first application that I know of that applies them to microwave data over an ice sheet. This is part of the novelty of the technique.**

**Revised text.**

**L36-40 We use an empirical algorithm previously developed to map the extent of Greenland's perennial firn aquifers via fitting exponentially decreasing temporal L-band signatures to a set of sigmoidal curves. This algorithm is recalibrated to also map the extent of ice slab areas using airborne ice-penetrating radar surveys collected by NASA's Operation Ice Bridge (OIB) campaigns (2010-2017).**

*P1, l48-53: Manuscript has many of these extremely long sentences. Please tighten up.*

**Will revise to shorten long sentences.**

**Revised.**

*P2, l68: Where is the aquifer introduced? Explain what it is first.*

**There is a general description of what a firn aquifer is in both the first paragraph of the Abstract, and in the second and third paragraphs of the Introduction. Although more details are provided in the main text, firn aquifers can simply be described as 'subsurface meltwater reservoirs consisting of a meters-thick water-saturated firn layer'..., which is the first line of the abstract. For this paper, that is**

the key physical characteristic that we observe via SMAP. See next paragraph for further comments.

**L17-18** *Perennial firn aquifers are subsurface meltwater reservoirs consisting of a meters-thick water-saturated firn layer that can form on spatial scales as large as tens of kilometers.*

**L62-73 Similar to subglacial lakes, perennial firn aquifers also represent radiometrically cold subsurface meltwater reservoirs (Miller et al., 2020) consisting of a 4-25 m thick water-saturated firn layer (Koenig et al., 2014; Montgomery et al., 2017; Chu et al., 2018) that can form on spatial scales as large as tens of kilometers (Forster et al., 2014). Perennial firn aquifers have been identified via field expeditions (Forster et al., 2014), airborne ice-penetrating radar surveys (Miege et al., 2016), and satellite microwave sensors (Brangers et al., 2020; Miller et al., 2020) in the lower-elevation (<2000 m a.s.l.) percolation facies of the Greenland Ice Sheet (GrIS) at depths from between 1 m and 40 m beneath the ice sheet surface. They exist in areas that experience intense seasonal surface melting and rain (>650 mm w.e. yr$^{-1}$) during the melting season and high snow accumulation (>800 mm w.e. yr$^{-1}$) during the freezing season (Forster et al., 2014). High snow accumulation in perennial firn aquifer areas thermally insulates water-saturated firn layers from the cold atmosphere allowing seasonal meltwater to be stored in liquid form year-round if the overlying seasonal snow layer is sufficiently thick (Kuipers Munneke et al., 2014).**

*P2-3 general comment: First explain, in less words, what a aquifer, ice slab and perched aquifers are, then come to the techniques used to measure them. Now it's back and forth between the two. The structure of the introduction is not very logical.*

**The paper is focused on demonstrating the potential of L-band radiometry to map englacial hydrological features on the Greenland Ice Sheet. The introduction is specifically structured to discuss the current state of knowledge –what do we know? Not much. The very first observation that surface L-band brightness temperature is sensitive to deep subsurface meltwater (subglacial lakes) was by Jezek et al., (2015). There is one previous L-band microwave radiometry paper on shallower subsurface meltwater (firn aquifers) mapping by Miller et al. (2020). There are no previous L-band microwave radiometry papers on ice slabs. This is the first. We are not aware of any papers (or observations!) on mapping perched firn aquifers. It is hypothesized from the modeling exercise in this paper. The introduction is meant to describe known features of firn aquifers and ice slabs that are sensitive to L-band emissions, and the interactions that generate specific L-band signatures that are used to map them.**

*P2, l76: "…through winter".*

**See previous comments. Authors will include 'year-round' in the revisions.**

**See L71-73.**

*P3, l85-95: Too technical for introduction. Why is this here? Either remove or combine with paragraph p4, l119-128.*

**Will revise.**

**These paragraphs removed from introduction to the beginning of the methods section.  See L137-150.**

*P5, l 152-175: This paragraph is likely better placed at the beginning of the introduction, as it is good to start by encouraging the reader by noting what is so special and important about the aquifers, instead of concluding the introduction with this.*

**The authors will move this paragraph to an Implications section to shorten the introduction as advised by Reviewer #1.**

**This was shortened and moved to the implications section. See L647-658.**

*P6, l209-214: This is exactly some thing that should be in the introduction and not in the methods sections. Many things are actually repeated through the methods section, and could be removed to clean up the manuscript.*

**Although we feel that these two paragraphs are probably relevant, the authors will remove them to shorten the paper.**

**Paragraphs removed.**

*P38: 1116-1150: What a big blob of text. Try to at least introduce some indentations to improve readability. To me, this paragraph is unclear. What would you want to improve exactly? Try to subdivide the respective future topics more clearly.*

The authors will put in some additional indentations, subdivisions to help improve readability. Future work (now our current work) is focused on developing better mapping algorithms based on forward and inverse geophysical-electromagnetic modeling and a potential path forward. Forward modeling provides insight into what parameters might be controlling the observed L-band signatures. Inverse modeling allows us to use these insights to more accurately map firn aquifers and ice slabs on an ice-sheet scale.

**The Summary and Future Work section was shortened. See L687-729.**

*P38: What about applications on other ice sheets?*

**Thank you. This is a good suggestion. For this paper, the authors made the decision to focus exclusively on Greenland, and not to include any mention of Antarctica (the paper is too long anyway!). We a forthcoming paper that will**

**describe an algorithm to map firn aquifers on ice shelves + a field expedition to validate the algorithm.**

**Author Response Reviewer #3**

**Note: the editor initially asked the authors to respond to the reviewers' comments before revising the paper. So, the authors first described the intended revisions. The authors have included these original comments bolded in black. However, the final changes that were made to the manuscript are in red.**

*As usual, I wish to iterate that the authors have engaged in far more work and consideration in writing this paper than I have undertaken while reviewing it. My comments here are meant to be productive and lead to a better manuscript. If, however, any part of my review seems unfair, or if it misunderstands the authors' points (or even misses them entirely), then I fully encourage and welcome the authors to respond accordingly with rebuttals to those points. I hope that the process remains constructive.*

*Primary comments on scientific merit:*

*On the whole, this manuscript is a solid (liquid?) contribution to the ice sheet remote sensing community. The recent discoveries of two previously unmapped hydrologic regimes across Greenland's percolation zone, which already cover nearly 25% of the entire ice sheet as mapped by airborne radar, are each widespread hydrologic phenomena that need to be monitored. To my knowledge, this is the first paper that has proposed a repeatable method for monitoring these facies using spaceborne sensors. These methodologies equip the cryospheric community to learn far more as these facies (presumably) are poised to cover an increasingly large fraction of the Greenland (and Antarctic) ice sheet(s). This is an important contribution to science, and one that should absolutely be added to the canon and taken seriously for future studies of ice sheet hydrology.*

**Thank you for these positive comments. With collaborators, our intention is to do just that.**

**The authors want to again sincerely thank review #3. These were fantastic comments that resulted in the authors doing a great deal of additional thinking and work on the concept of 'perched' firn aquifers.**

*The idea of "perched firn aquifers" is an interesting one. Miller and co-authors are correct here in noting that interannual variability in melt & accumulation are very large in Greenland's percolation zone. It has been an open question about "what happens in areas where the accumulation thresholds for ice slabs vs firn aquifers fluctuate between these thresholds from year to year, or every few years?*

*The hypothesis that aquifers could form occasionally on top of ice slabs, as proposed here in an "intermittent perched aquifers" hypothesis, would rely on a few assumptions about the meteorology at such a firn column.*

*(1) That enough meltwater was produced in a given year to stay liquid (not fully freeze) in the oncoming winter;*

*(2) that meltwater would be able to get deep enough (even atop an ice slab) to be insulated by overlying layers winter snow & firn (existing documented firn aquifers have all been at least 5-20+ meters under the surface at their tops); and*

*(3) that enough winter accumulation would form to adequately insulate that water-saturated firn against the winter cold.*

*It is worth noting that no accuracy analyses (false-positive and -negative) have been performed on the Icebridge-derived radar maps of aquifers and ice slabs produced by Miege et al and MacFerrin et al, respectively. (This isn't inherently a criticism of their work… having made the first observational maps of each respective facies, it is unclear what "truth" they could have been compared against save for a few individual field cores.). Miege, et al (2016) had no knowledge of the existence of ice slabs identified & mapped in these regions by MacFerrin, et al (2019). MacFerrin, et al (2019) identified more than 100 buried lakes on top of ice slabs that were filtered out in their analysis. These are many of the same buried lakes identified several years earlier by Koenig et al (2015), mapped using the same radar instruments. Miege, et al (2016) identified a number of these regions as "aquifers", even though they do not necessarily meet the definition of permanently soaked porous medium. (Miege would not necessarily have been able to differentiate between buried lakes and aquifers, as the radar signatures alone are essentially identical once the signal hits deep water.).*

*Koenig, et al., 2015 do document buried perennial supraglacial lakes, but those are a different (and more spatially-limited & localized) phenomenon than what is proposed here as a perched aquifer An individual buried lake is not inherently an aquifer and doesn't require the same thresholds of surface-mass-balance conditions (high melt, very-high accumulation) to sustain itself, given that water flows horizontally over great distances into localized lake basins and reaches local water depths far greater than possible by vertical percolation alone.*

**The authors started writing this response by looking for additional data that would support our classification and mappings. We analyzed 'buried lake' locations derived by Koenig et al., (2015) between 2009 and 2012 using OIB data, and by Dunmire et al., (2021) between 2018 and 2019 using Sentinel-1 SAR data, visible satellite imagery, and RACMOp2.3 climate simulations together with our results. During our analysis, the authors came to several conclusions and a stronger hypothesis, and will make the following revisions to the manuscript.**

**(1) The term 'perched firn aquifer' is too simple. We will change this terminology to 'other englacial hydrological features'. In the percolation facies during the freezing season, englacial hydrological features likely range from deep, expansive, perennial firn aquifers that form in firn pore space, to small, shallow, supraglacial lakes that form on relatively impermeable ice.**

**(2)** These features form intermittently, often in ice slab areas, likely due to these relatively impermeable layers buried within the firn. However, they can also exist anywhere above, below, or within the snow, firn and ice, regardless of local climate conditions as a result of vertical meltwater percolation combined with lateral meltwater flow.

**(3)** Meltwater is retained on variable time scales (weeks-months, years) during the freeing season.

**(4)** The effective resolution of SMAP (~18 km) is too coarse, and the sampling of the airborne radar data too sparse to effectively calibrate our current empirical model and accurately classify and map 'other englacial features', although we hypothesize they exist and can be observed in the L-band microwave signatures. The authors will remove the 'perched firn aquifer' classification from our maps, and simply map firn aquifers and ice slabs.

**(5) The authors will add a discussion on the potential controls on the decreasing 2-year elevated-but-declining plateau of L-band TB, which will include 'buried lakes' maps, visible satellite imagery, and RACMOP2.3 climate simulations as supporting lines of evidence. The authors do strongly believe that the signatures represent subsurface meltwater, but we will also present several alternate hypotheses. We will discuss future work using this L-band signature to develop an an algorithm that does not require calibration with airborne radar data.**

**The authors did a significant amount of work as stated above as well as additional electromagnetic modeling of the L-band signatures. These signatures exist across ice slab areas in the percolation facies in all years of the SMAP data. We firmly believe that these signatures represent subsurface meltwater, and can be used for future mapping. Indeed – my young co-author Riley has a very cool airborne paper in progress that will identify the physical characteristics of these areas. However, the authors made the decision to remove 'perched firn aquifers' from this paper for the following reasons:**

**(1) The paper was already way too long, and the additional modeling and discussion would have made it longer.**
**(2) This paper was designed as an empirical study. We do not (yet) have airborne observations that provide the locations and a description of the physical characteristics of 'perched firn aquifer' areas. The analysis of the microwave signatures would need to be presented in terms of an electromagnetic model.**
**(3) This concept needs to be further developed, which will take time.**

**The authors made the decision to keep the paper as a strictly empirical study on perennial firn aquifers and ice slabs – and move this manuscript forward. The authors are planning a second manuscript detailing the empirical algorithm+**

**electromagnetic modeling on the concept on 'perched firn aquifers' once the airborne paper is submitted. Reviewer #3 was bang-on with their comments.**

*The authors propose that a temporary perched aquifer formed near the upper portion of the K-transect area and then retained some amount of liquid for the next two years before fully freezing up, is at least partially supported by the presence of an anomalously elevated but gradually-decreasing 2-year apparent trend in L-band TB spanning from summer 2016 through 2018, as shown in figure 4b. And indeed, as seen in that same panel, 2016 was a significantly higher melt year than any of the other three years in that record, potentially meeting condition.*

*(1). But the IceBridge AR radar profiles shown in Figure 3b show the upper horizon of ice slabs very near the surface, within the top 1-2 m of snow & firn (these profiles were collected in April & May, at the end of the winter season when snow is at its deepest of the annual cycle). The bottom of a perched aquifer there would not be anywhere near the closest upper-horizon of deep aquifers seen in any existing documented literature.*

*(2), the annual accumulation in that region of southwest Greenland, partially rain shadowed by the Maniitsoq ice cap, is typically 0.2 – 0.6 m w.e., or perhaps 0.6-1.5 m of snow. It would require a substantial amount of additional snow to insulate a perched water layer in this region (Kuipers-Munneke, et al, [2014] show this with modeling, low accumulation regions do not physically form aquifers). That alone does not disprove their presence though. One would hope that, perhaps in the way that both aquifers and ice slabs were originally discovered by in-situ cores (Forster, et al, 2014, and Machguth, et al, 2016, respectively), that perhaps some in situ data could be found to document such a phenomenon.*

**Reviewer #3 is correct in pointing out that there is no perched firn aquifer in the Icebridge echogram (Fig. 3b) where the TB time series (Fig. 4b) is discussed.**

**The perched notation**

**The authors agree that modeling does not tell the complete story – as perennial firn aquifers and ice slabs were not explicitly predicted by models prior to their discovery.**

**It is important to first note the significant difference in the scale of the satellite footprint vs the airborne footprint. The gridding of the TB observations used to derive the maps is 3.125 km; however, the effective resolution is ~18 km. The gridding (or trace spacing) of the IceBridge echograms is15 m x 20 m (Accumulation Radar) or 14 m x 40 m (MCoRDS). For a full 18 km airborne transect across the satellite footprint, the complete IceBridge echogram observes 2-3% of what the satellite observes. In other words, 97-98% of what the satellite observes is unknown. The authors note that there are mapped 'buried lakes' within the satellite footprint of the Icebridge echogram (Fig. 3b), and observable lakes and lateral drainage off ice slabs in visible satellite imagery.**

*Fortunately, there were multiple field teams in that region of Greenland in 2017, namely the GreenTRACs campaign who collected in situ radar in that immediate vicinity, and the FirnCover campaign who collected at least one core at KAN-U in April 2017, in the immediate vicinity of that cyan dot in Figure 2b (MacFerrin, et al, 2019, Figure S1.b). Neither in situ campaign showed any evidence of liquid water perched shallowly above an ice slab in that region in Spring 2017, nor do they show the anomalously high snow accumulation rates that would be needed to do so. The MacFerrin, et al, 2019 core from KAN_U shows no water at all. The AR profile shown in Figure 3b in this manuscript, also does not indicate an anomalously bright reflector followed by a near-complete signal-extinction to depth as would be characteristic of a liquid water table (seen in Figure 3a in Southeast Greenland), nor the high (2-3 m thick) annual snow layer atop the ice slabs that would be needed to insulate it. If such an aquifer water table existed there at that time, one would expect to see it in that airborne AR transect, right where the cyan circle is placed in Figure 2b, but I simply don't see any evidence of liquid water there.*

**We note that a firn core is on the centimeter-scale as compared to the ~18 km effective resolution of the satellite, which significantly less than meter-scale of the airborne transect. It is possible that a single core could 'miss' a diffuse liquid layer especially if that layer is intermittent in time.**

*Comment: since conclusions are being made about specific points on the ice sheet in Figure 2, the exact locations (lat/lon) of the icons shown in Figure 2 need to provided, especially since a new firn hydrology regime is being proposed at one of those specific points. (If they are, I did not see them and apologize for the oversight.)*

**Thank you. Very good suggestion. The authors will add these lat/lon points to the text/figure or in a related table depending on the number of points and their relevant metadata.**

*I cannot (and won't) completely rule out that "intermittent perched aquifers" could possibly exist under the right firn conditions, but I don't see the multiple lines of evidence necessary to make such a conclusion yet, nor a piece of irrefutable evidence (such as discovery of such a hydrologic regime in a set of firn cores or snow pits). To be fully clear, I am not wholly sure what else would cause a 2-year elevated-but-declining plateau of L-band TB values seen in Figure 4b. In fact, I am not entirely sure it truly is a "2 year" trend as hinted by the drawn arrow, it could just as likely be two non-related seasonal one-year declines (nearly identical to every other year at that location) but with 2016 in particular seeing higher values than other years for some reason. Given that the IceBridge AR data presented do not support the existence of an aquifer there, and adjacent cores by other campaigns/papers in that region in that Spring don't as well, and anomalously high accumulation rates necessary for such an aquifer to be properly insulated don't seem to be present there either, I cannot really support a claim of the discovery of "intermittent perched aquifers" in this manuscript based upon L-band TB signatures alone. More evidence, or more concrete and non-conflicting evidence, would be needed to make that claim. I do welcome the authors to broaden the discussion*

*about what could have caused such a jump in annual TB values at that location in 2016-17, such as winter snow accumulation differences, the refreezing of anomalously-high amounts of water there, or other possible causes. Perhaps even include an adjacent discussion of a possible "perched firn aquifer" (although again, they would need to explain why it isn't seen in Spring 2017 airborne radar data), and suggest it as a possible explanation. But going straight to the conclusion that intermittent perched aquifers have now been discovered, and subsequently mapped, I do not believe is strongly enough supported solely by an interpretation of L-band TB microwave signals with no other supporting evidence (and multiple lines of contradictory evidence).*

**We agree with the reviewer and will add qualification to the text to more clearly state that this is a difficult hydrologic regime (perhaps multiple hydrological regimes based on location and climate), and that while we strongly feel it likely based on the observed L-band signatures, it cannot be mapped without significant uncertainty on the basis of the available remote sensing to date. We plan to add similar language to the discussion to encourage further work in the field.**

**The authors note that they don't consider a satellite mapping a discovery per say, but rather a prediction of what may be hiding underneath the surface. Brightness temperature signatures are too complex (and often non-unique) to definitively define a new hydrologic regime without multiple lines of evidence.**

**A satellite prediction map might be the first step in the discovery of such phenomenon. Indeed, it was the authors prediction maps that led to the discovery of an expansive perennial firn aquifer on the Wilkins Ice Shelf, Antarctic Peninsula via firn cores, GPR, and airborne radar surveys during a 2020 expedition (Miller et al., 2019; https://ui.adsabs.harvard.edu/abs/2019AGUFM.C33A.07M/abstract). Once the phenomenon was identified, multiple lines of evidence indicated that these features had likely been present, yet undetected, for decades (since at least the early 1960s).**

*(Final note: my conclusion about the perched aquifers is not irrefutable. As I noted at the start, if I have missed important aspects of the analysis, or there is more evidence to support the claims made of a new "perched aquifer" ice sheet regime that I do not see, such as firn model data that may suggest and alternating regime between ice slabs and aquifers, and the evidence contradicting it can be refuted or explained, then I would love to stand corrected. It would be an exciting new development in ice sheet firn hydrology.)*

**Game on. 😉**

**Game on… in the next paper. 😊**

*Again, it may be possible that some form of "perched aquifer" exists in Greenland, but the map provided here does not inherently lead to that conclusion without further*

*evidence. However, none of this detracts from the fact that using a single L-band sensor to reliably map both aquifers and ice slabs across the Greenland ice sheet is a significant scientific advancement, and more than worthy of publication. I suspect, if revised, accepted, & published, this work is likely to become a seminal contribution to future methodologies for the remote sensing of polar ice sheets. For that reason alone, I truly hope the authors can adjust their interpretations noted above, and get the manuscript accepted and published.*

*Overall the manuscript is well-written and readable, and the figures and maps are generally helpful and understandable.*

**Thank you again for these positive comments. They are sincerely appreciated by the authors.**

*Minor comments and grammar edits:*

*Lines 48-53: "As Greenland's climate continues to warm… …and stability of the Greenland ice sheet." This is a very long sentence. Consider breaking into two.*

**Thank you. The authors will break this into two sentences in the revised manuscript.**

**L43-48 Revised. Text below.**

***As Greenland's climate continues to warm, seasonal surface melting will increase in extent, intensity, and duration. Quantifying the possible rapid expansion of these sub-facies using satellite L-band microwave radiometry has significant implications for understanding ice sheet-wide variability in englacial firn hydrology resulting in meltwater-induced hydrofracturing and accelerated ice flow as well as high-elevation meltwater run-off that can impact the mass balance and stability of the GrIS.***

*Line 63: "Recent launch of … has provided…" --> "The recent launches of … have provided…"*

**Thank you. This will be corrected.**

**L 50-53. Good catch. Corrected. Text below.**

***The recent launches of several satellite L-band microwave radiometry missions by NASA (Aquarius mission, Levine, et al., 2007; Soil Moisture Active Passive (SMAP) mission, Entekhabi et al., 2010) and ESA (Soil Moisture and Ocean Salinity (SMOS), Kerr et al., 2010) have provided a new Earth-observation tool capable of detecting meltwater stored tens of meters to kilometers beneath the ice sheet surface.***

*Line 131: McFerrin, et al., 2019) --> MacFerrin, et al., 2019) (also correct in other locations where this reference is used).*

**Will be corrected.**

**All MacFerrin references corrected.**

*Lines 135-137: "Particularly in areas that experience intense seasonal surface melting (>600 mm yr−1) during the melting season, and lower snow accumulation (<600 mm yr−1) during the freezing season as compared to perennial firn aquifer areas (McFerrin et al., 2019)."It's unclear if the units being used here are millimeters water-equivalent, millimeters snow-equivalent, or ice-equivalent (which differ by up to a factor of 3). Please clarify. If mm w.e. are used, MacFerrin et al used somewhat different numbers than this in their empirical analysis (266-573 mm w.e. yr-1 for melt, not >600 mm yr-1). The estimate of snow accumulation (572 +/- 32 mm w.e.) used by MacFerrin, et al., does overlap with what is cited here (<600 mm yr-1).*

*Also, the sentence quoted above is a fragment; fix grammar to complete the sentence.*

**Thank you. The units and grammar will be corrected.**

**L119-120. Thank you again.**

**They exist in areas that experience intense seasonal surface melting and rain (266-573 mm w.e. yr−1) during the melting season, and lower snow accumulation (<572+/-32 mm w.e. yr−1) during the freezing season as compared to perennial firn aquifer areas (MacFerrin et al., 2019).**

*Line 144: "This results in a lower observed TB at the ice sheet surface during the freezing season." It is unclear what "lower observed TB at the ice sheet surface" in this sentence is compared to, as the previous statement compares ice slabs both to aquifers and other non-ice-slab facies. Please clarify.*

**Will be corrected.**

**L126-131. Corrected. Added text below.**

**This results in typically higher observed $T^B$ at the ice sheet surface during the freezing season in ice slab areas, as compared to other percolation facies areas, however, typically lower observed $T^B$ as compared to perennial firn aquifer areas. Similar to temporal L-band signatures over perennial firn aquifer areas, temporal L-band signatures over ice slab areas are exponentially decreasing during the freezing season, however, the rate of $T^B$ decrease is slightly more rapid.**

*Lines 212-214:"Critically, the majority of meltwater is stored at depths that only L-band satellite microwave sensors (i.e., radiometers, radar scatterometers, and synthetic aperture radars) are capable of detecting."*

*Should specify: L-band microwave sensors are the only known category of space-borne instruments that are presently known to be able to detect water at these depths. We do not want to infer that no other instrument could ever exist that would do this. (For instance, we also know in situ active seismic sensors can detect aquifer depth quite well, but of course none of us know what a spaceborne seismic sensor would even look like, I shudder at the thought.) But for now just stating that L-band is the only category of sensors currently proven to detect these features, suffices.*

**This is a great, thoughtful comment, and I agree, and would have made a stronger comment to that effect but moderated in order to let the data speak for itself.**

**These lines were removed from the revised text to shorten the manuscript.**

*Lines 286-296: I am glad to see the acknowledgement of sources of uncertainty given the slight temporal mis-matches of datasets here. I do not propose a method to eliminate these biases, as I am not sure that is possible given the data available. I am just commenting that this section appears well written and considered, and I am glad the authors included it here.*

**Thank you for the positive comment. The authors do not believe these biases can be eliminated either. The temporal mis-matches as well as the spatial mis-mactches between the airborne and satellite data are impossible to overcome from the available data, especially given IceBridge has ceased its regular operations.**

*Line 371: Culberg et al., 2021) → Culberg et al. (2021) (fix parentheses)*

**Thank you. Will be corrected.**

**Reference corrected.**

*Line 374 (Figure 3): The color-bars (along the right axes) in panels (a) and (b) lack units/labels. Please add.*

**Will be corrected.**

**This figure was revised, and the issues was corrected.**

*Lines 452-455 (Table 1): "Coverage (km2)" is used as an identical header on two columns here. It is identified in the caption that one is the coverage of detections in rSIR grid cells, and the second is the detections that overlapped AR-derived detections, but this should be made more clear in the column titles somehow.*

**Will be corrected.**

**This table was removed to shorten the manuscript. The values are instead within the text.**

*Line 467:"and more recent studies using L-band microwave radiometry" … two lines above, you define the frequency ranges used for Ku-band and P-band, but I don't see anywhere you define the frequency ranged referred to as "L-band" here. I saw you define the wavelength further above in the paper [21 cm] but not the frequency bands. (I may have just missed it, apologies if I did.) Listing the L-band frequency would be helpful to non-microwave experts, just for comparison.*

**As suggested by Reviewer #1, this text will be removed to shorten the manuscript.**

**This paragraph was removed.**

*Line 477:"Tedesco and Fettweis, 2020" is listed twice. Remove one.*

**Thank you, this will be corrected.**

**Thank you, again. Corrected.**

*Line 942:"Overlapping perennial firn aquifer and ice slab detections are interpreted as perched firn aquifer areas."*

*This is, perhaps, the source of some of the disagreement in the lengthy commentary I gave above*

**The authors will remove the perched firn aquifer classification from the empirical model description as well as the mapping. We will instead include only perennial firn aquifers and ice slabs.**

**All reference to 'perched firn aquifers' were removed from this paper.**

---

## Author Response (AR2)

**Suggestions for revision or reasons for rejection (will be published if the paper is accepted for final publication).**

I would like to congratulate the authors on a very fine piece of work. This paper is a significant advancement in our collective ability to remotely sense the subsurface percolation facies of the Greenland ice sheet, It takes a significant leap above other L-band-based studies (including recent works by the authors) that calculated only the extent of firn aquifers across Greenland. The inclusion of ice slabs, using a single instrument, gives us the "other half" of the meltwater equation, and unlocks the future possibility of remotely sensing these facies annually to study the evolution of the ice sheet's percolation facies over time in a warming climate.

**The authors thank the reviewer for the positive comments.**

This manuscript has made significant improvements over the original manuscript, not the least of which is the clarity of its goals and conclusions. The simplification of the algorithm to detect "known" facies that have been previously mapped (even if only incompletely from airborne radar) represents a substantial advancement, without teetering into speculation of the unknown or unproven. This simplification does come with drawbacks, and the authors note that their algorithm as presented here does not have the ability to determine interannual changes in aquifer or ice-slab extent. But as they note, that possibility still exists with potential future work and further analyses.

**Slight correction: we are definitely able to map interannual variability; however, with the uncertainty described in the paper**.

I have some questions and a bit of a raised brow about some of the results outlined in Figure 7, with ice slabs & aquifers appearing in some regions where they have distinctly *not* been picked up using any other methods before. I note specifically the ice slab regions in far South Greenland (between 60-62.5 N) where no ice slabs have ever been found, as well as some isolated aquifers detected by this method in far North Greenland (above 78 N) where accumulation rates are almost certainly far too low to geophysically produce such perennial aquifers. Those areas raise a bit of concern about the reliability of the algorithm. But, I think the authors have taken care in this manuscript to identify this product as a "draft" that is subject to further revisions as time and effort allow, and they have deliberately noted the fact that even slight changes in the cutoff parameters can cause large shifts in the boundaries between these facies. Since I have no suggestions to improve their algorithm to fix these anomalies, and the authors note that errors likely do exist due to very-slight adjustments in the algorithm's parameters, I do not see a need to withhold the publication of this paper any further based on those uncertainties. They stand as they are and the authors note the uncertainties within. (And, perhaps, some future fieldwork or airborne campaign might discover these facies in those areas... I have to remember as a reviewer that the absence of evidence does not imply evidence of absence. The authors make a plausible case for why these facies may exist there despite never having been found there. Since I can't prove these facies *don't* exist in every spot they are identified here, it is sufficient to let the algorithm pass as is and save refinements for another day.)

**The authors tried to be very specific about the limitations of the algorithm, and associated uncertainty. This is the limit of the four-parameter ice sheet-wide single frequency (L-band) mapping algorithm. It simply can't be tuned any closer to the airborne data, and that may not be appropriate because of the time shift between the two data sets. The authors are currently working on an active-passive multi-frequency approach which may reduce the uncertainty. But, it is a complicated retrieval.**

This manuscript reads very well. Much of the methods are heavily technical in nature, but this is a methodology paper and the techniques used to process the data must be adequately described. (I disagree wholeheartedly with one of the other reviewers' comments on the original manuscript submission, who called the paper "too technical"... it would be incomplete if the authors left out most the equations and only vaguely described their methodologies using hand-waving descriptions. But I digress.)

**Thank You. The authors are very much in agreement that the technical details are critical to this particular manuscript, and provide a more rigorous foundation to step from a**

**technical paper to a science paper – where the obvious objective will be to identify interannual variability and potential expansion.**

The inclusion of "spatially-coherent melt layers" is perhaps unnecessary in this manuscript, given that they are not being detected or mapped at all by the SMAP algorithm, and seem a bit extraneous to the conclusions of the paper. However, their inclusion in Figure 7b as well as the zoomed figures 8 & 9 do help tell an interesting story, and as such I don't think they really need to be omitted for the manuscript to be approved. They are just a bit unnecessary to make the conclusions in the study, but don't need to be removed, in my opinion.

**The authors agree that the "spatially-coherent melt layer" inclusion is a bit tangent to the main results of the mapping algorithm. The key reasons the authors decided to keep them in the manuscript were (1) they were a pronounced feature in the radargram in Fig. 3a, (2) there are distinct in L-band signatures throughout the upper percolation and dry snow facies (i.e., Figure 4d), (3) the authors believe in the future they will play a significant role in the evolution of both perennial firn aquifer and ice slabs (Section 4), and (4) there is clear potential to develop an L-band algorithm to map these features.**

I see no significant reason (save one) for publication of this manuscript to wait any longer. (I outline the one caveat at the end of this review.) Let it be published. I have very slight line-item recommendations to make, outlined below. Not really "revisions", per se, just grammar checks, really, as well as some basic minutae. The authors can likely address most of these in 15 minutes or less.

Line 135: Include a period (".") at the end of this paragraph.

**Corrected.**

Line 156: "and an innovative extension of the science objectives." Perhaps for clarity, add the word "mission's": "and an innovative extension of the mission's science objectives." (Up to you, though.)

**Corrected.**

Line 275: "relatively thin (0.02 cm-2m)" ; Did the authors mean 0.2 cm (2 mm) rather than 0.02 cm? I don't know that anyone has been able to measure ice lenses just 0.02 cm (0.2 mm, just 200 microns) thick. Easy typo.

**Are you sure we can't measure ice lenses that thin? 😣 Corrected.**

Line 361: I'm not sure that "runs-off" needs a hyphen there. Perhaps remove.

**Corrected.**

Line 636: "McFerrin" --> "MacFerrin"

**Corrected**

Line 672-673: "...meltwater run-off across ice slabs in the percolation facies was recently observed in visible satellite imagery collected by the NASA-USGS Landsat 7 mission during the 2012 melting season (MacFerrin et al., 2019)" ; Although MacFerrin, et al (2019) did prominently note this runoff over ice slabs in Figure 1 of their 2019 paper, the first use of this particular image was in Machguth, et al (2016), which displayed the same Landsat-7 image (with slightly different processing, used by an overlapping list of authors) to first illustrate runoff over impermeable ice layers. I would suggest changing this particular citation to "Macguth, et al (2016)" to note the first time this was observed in satellite imagery.

**Thank you for noting the mistake. Corrected.**

Lines 713-718: This paragraph (starting with "Future work will focus on...") reads more like a research proposal than

a paper conclusion. This work may get done (I certainly have full confidence in the authors' abilities to do what they describe), but it seems unnecessary to list out future plans in the conclusion of a research paper. It's up to you and the editors, but I might consider omitting this paragraph as just being somewhat unnecessary. That isn't a "demand" though, if the authors & editor agree it's stronger with this paragraph in there, leave it be.

And then, the last point:

Line 750: "[data] are available from JZM upon request." I was excited about this manuscript right up until this line. 100% of this work was publicly funded, and the manuscript is being published in an open-access journal. "Data available [only] upon request" was a relict of the 20th century that needs to go away. In the entire paragraph preceding that statement starting on line 731, the authors did not need to personally 'beg' every lead author of the 10 different respective data products they used; it would have been ungainly and prohibitive to do so. The authors were able to publicly download every one of those datasets freely. This manuscript wouldn't have been possible without access to that data. Keeping the data presented in this paper behind a "locked door" is no longer an ethical use of resources, and in fact runs afoul of the stated policies of the NASA programs (see Acknowledgements) that funded this work. This one detail is the reason I had to change the conclusion from "Accepted as is" --> "Accepted subject to minor revisions." Publicly-funded work, built entirely using publicly-available data by scientists working in publicly-funded salaried positions, published to publicly-accessible open access journal, should be made public.

**Whoah! Ease Back!**

**The authors are in complete agreement that publicly funded data should be publicly available. There was never any intention by the authors to keep the data behind a 'locked door' or make anyone 'beg' for it. Indeed, the data has already been very freely passed to other scientists for analysis prior to publication, with an open invitation for discussion, especially given the significant uncertainty noted in the above paragraphs.**

**The algorithm and the data are being actively developed as part of a SMAP Science Team project, which includes public distribution of the data (and metadata) through the NSIDC DAAC together with a full Algorithm Theoretical Basis Document (ATBD). Since this is an early version – a prototype - the author (JZM) was simply trying to make sure the data distributed to the public were the most up to date version at the time of the request.**

**That data has been uploaded here: https://www.scp.byu.edu/data/aquifer/.**

Other than that, great paper. Excellent revisions. Let's see it get out!

---

## Author Response (AR3)

Lars –

Thank you for the suggestion for the data repository. The data has been uploaded and now has a DOI, consistent with The Cryosphere's data policy.

~JZ